# ASTRA: A Scene-aware TRAnsformer-based model for trajectory prediction

**Izzeddin Teeti**                                                              *izzeddin.teeti@gmail.com*
*Visual Artificial Intelligence Laboratory (VAIL), Oxford Brookes University*

**Aniket Thomas**[*]                                                           *aniket.thomas@iitb.ac.in*
*Indian Institute of Technology Bombay*

**Munish Monga**[*]                                                           *munish30monga@gmail.com*
*Indian Institute of Technology Bombay*

**Sachin Kumar**[*]                                                                    *sgiroh@gmail.com*
*Indian Institute of Technology Bombay*

**Uddeshya Singh**[*]                                                          *ud.uddeshya16@gmail.com*
*Indian Institute of Technology Bombay*

**Andrew Bradley**                                                             *abradley@brookes.ac.uk*
*Oxford Brookes University*

**Biplab Banerjee**                                                              *getbiplab@gmail.com*
*Center of Machine Intelligence & Data Science, Indian Institute of Technology Bombay*

**Fabio Cuzzolin**                                                         *fabio.cuzzolin@brookes.ac.uk*
*Visual Artificial Intelligence Laboratory (VAIL), Oxford Brookes University*

**Reviewed on OpenReview:**

## Abstract

We present ASTRA (**A S**cene-aware **TRA**nsformer-based model for trajectory prediction), a light-weight pedestrian trajectory forecasting model that integrates the scene context, spatial dynamics, social inter-agent interactions and temporal progressions for precise forecasting. We utilised a U-Net-based feature extractor, via its latent vector representation, to capture scene representations and a graph-aware transformer encoder for capturing social interactions. These components are integrated to learn an agent-scene aware embedding, enabling the model to learn spatial dynamics and forecast the future trajectory of pedestrians. The model is designed to produce both deterministic and stochastic outcomes, with the stochastic predictions being generated by incorporating a Conditional Variational Auto-Encoder (CVAE). ASTRA also proposes a simple yet effective weighted penalty loss function, which helps to yield predictions that outperform a wide array of state-of-the-art deterministic and generative models. ASTRA demonstrates an average improvement of 27%/10% in deterministic/stochastic settings on the ETH-UCY dataset, and 26% improvement on the PIE dataset, respectively, along with seven times fewer parameters than the existing state-of-the-art model (see Figure 1). Additionally, the model's versatility allows it to generalize across different perspectives, such as Bird's Eye View (BEV) and Ego-Vehicle View (EVV).

---

[*]Authors are listed in alphabetical order and contributed equally.

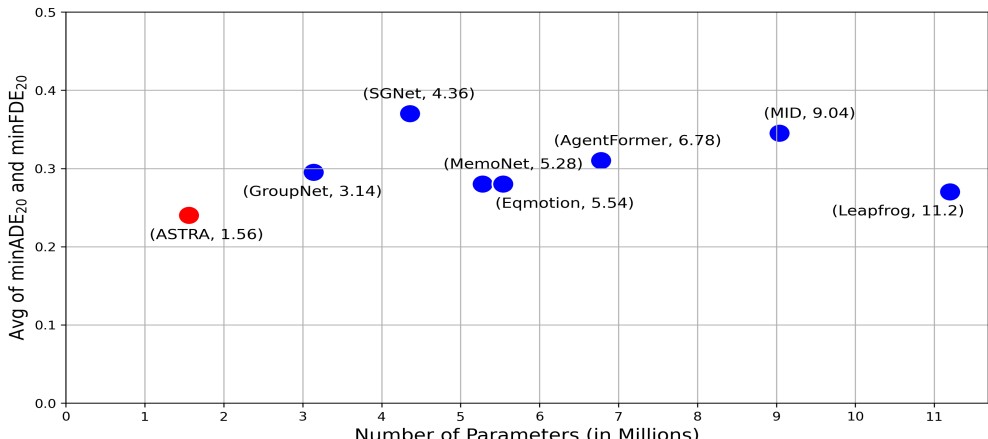

Figure 1: Comparison of average $(\text{minADE}_{20}/\text{minFDE}_{20})$ against the number of parameters for various models on the ETH-UCY dataset. Each point represents a different model, with the model name and number of parameters in millions indicated. Our model, ASTRA, achieves state-of-the-art results with the least number of parameters, demonstrating its efficiency and effectiveness in pedestrian trajectory forecasting.

# 1 Introduction

**Motivation & Importance** The pursuit of forecasting human trajectories is central, acting as a cornerstone for devising secure and interactive autonomous systems across various sectors. This endeavour is crucial in a wide array of applications, encompassing autonomous vehicles, drones, surveillance, human-robot interaction, and social robotics. Furthermore, it is crucial for predictive models to strike a balance between accuracy, dependability, and computational efficiency, given the imperative for these models to function on in-vehicle processing units with limited capabilities. The challenge of trajectory prediction involves estimating the future locations of agents within a scene, given its past trajectory. This estimation task can be tackled either through Bird's Eye View (BEV) or Ego-Vehicle View (EVV) perspectives. This demands a comprehensive understanding of the scene, in addition to spatial, temporal, and social aspects that govern human movement and interaction.

**Limitations in Existing Models** To solve the prediction problem, various building blocks, including RNNs, 3D-CNNs, and transformers, have been employed to address the temporal dimension, with transformers demonstrating superior efficacy Giuliari et al. (2021a); Rasouli & Kotseruba (2023). However, temporal modelling alone is unaware of the social behaviour of the agents within the scene, i.e. how agents interact with one another. In addressing the social dimension, methods such as Social Pooling Pang et al. (2021) and Graph Neural Networks Kipf & Welling (2016) (GNNs) have been explored, with GNN emerging as the most effective Xu et al. (2023). Some researchers have integrated both transformers and GNN, either sequentially or in parallel, to refine the prediction paradigm Li et al. (2022); Chen et al. (2023); Jia et al. (2023); Liu et al. (2022). However, these approaches entail heightened computational burdens due to the resource-intensive nature of both GNNs and transformers. Furthermore, transformers, by their inherent design, may pose potential challenges in preserving information, as they do not inherently accommodate the graph structure in their input. On the other hand, scene dimension, or scene embedding, delves into the interaction between an agent and its surroundings. Rasouli et al. (2021) and Mangalam et al. (2021) utilised semantic segmentation maps which enhanced the model's grasp of environmental context. Another aspect across all surveyed papers, however, is their tendency to focus exclusively on either BEV or EVV, rarely considering both like Yao et al. (2021). This narrow focus becomes particularly problematic in, e.g., unstructured environments where a BEV might not be available, limiting the applicability of these methods.

**Contributions of ASTRA** In light of these challenges, this paper introduces a lightweight model, coined ASTRA (**A S**cene-aware **TRA**nsformer-based model for trajectory prediction). By integrating a U-Net-

based key-point extractor Ribera et al. (2019), ASTRA captures essential scene features without relying on explicit segmentation map annotations and alleviates the data requirements and preprocessing efforts highlighted earlier. This method also synergises the strengths of GNNs in representing the social dimension of the problem and of transformers in encoding its temporal dimension. Crucially, our approach processes spatial, temporal, and social dimensions concurrently, by embedding the graph structure into the token's sequence prior to the attention mechanism, rendering the transformer graph-aware. The model does so while keeping the complexity of the model minimal. To refine the model's ability to accurately learn trajectories, we implemented a modified version of the trajectory prediction loss, incorporating a penalty component (detailed in Section 4.5). This is in contrast to Yuan et al. (2021) which does not build a graph and does not preserve the social structure; they distinguish between self-agent and all other agents, then they treat all other agents the same without encoding the positional or structural encodings.

Furthermore, distinct from the vast majority of models in this domain, our model demonstrates generalisability by being applicable to both types of trajectory prediction datasets, BEV and EVV.

Our methodology underwent evaluation using renowned benchmark trajectory prediction datasets ETH Pellegrini et al. (2009a), UCY Lerner et al. (2007), and the PIE dataset Rasouli et al. (2019). The empirical findings highlight ASTRA's outperforming the latest state-of-the-art methodologies. Notably, our method showcased significant improvements of 27% on the deterministic and 10% on the stochastic settings of the ETH and UCY datasets and 26% on PIE.

While maintaining high accuracy, ASTRA also features a significant reduction in the number of model's parameters (Figure 1), FLOPs (Floating Point Operations), inference running time & MACs (Multiply-Accumulate Operations) than the existing competing state-of-the-art model Mao et al. (2023); Yuan et al. (2021).

The paper's highlights are as follows:

1. A lightweight model architecture that is seven times lighter than the existing state-of-the-art model, tailored for deployment on devices with limited processing capabilities while maintaining state-of-the-art predictive performance.

2. A loss-penalization strategy that enhances trajectory prediction, featuring a weighting trajectory loss function that dynamically adjusts penalty progression in response to prediction challenges.

3. Utilisation of the Scene-aware embeddings with a U-Net-based feature extractor to encode scene representations from frames, addressing a critical aspect often overlooked in recent works.

4. A graph-aware transformer encoder that contributes to generating Agent-Scene aware embedding for improved prediction accuracy, ensuring informed inter-agent interaction capture.

**Paper Organisation** The paper is structured as follows: Section 2 reviews related work in trajectory prediction. Sections 3 and 4 defines the problem and details ASTRA's architecture and key innovations, respectively. Section 5 describes the experimental setup. Section 6 presents empirical findings and performance comparisons. Finally, Section 7 summarizes contributions, limitations, and future directions.

## 2 Related Work

### 2.1 Stochastic vs. Deterministic Approaches in Trajectory Prediction

The trajectory prediction problem is usually addressed in two ways: stochastic predictions Mao et al. (2023); Yuan et al. (2021); Xu et al. (2022b); Yao et al. (2021) and deterministic predictions Helbing & Molnar (1995); Pellegrini et al. (2009b); Alahi et al. (2016). A deterministic approach assumes a fixed future and produces only a single, most probable trajectory per input motion, making it suitable for scenarios where uncertainty is minimal or a single best estimate is required. In contrast, a stochastic approach acknowledges uncertainty in future motion and generates multiple possible trajectories for the same input, capturing the inherent variability in human movement. Stochastic approaches utilizes generative techniques like Conditional

Variational Auto-Encoders (CVAEs) Yuan et al. (2021); Yao et al. (2021), Generative Adversarial Networks (GANs) Huang et al. (2021), Normalizing Flows Bhattacharyya et al. (2020), or Denoising Diffusion Probabilistic Models Mao et al. (2023) to introduce randomness into the prediction process, thereby generating diverse future trajectories that better represent the distribution of possible pedestrian movements. ASTRA, like some prior works Xu et al. (2023); Yao et al. (2021); Salzmann et al. (2020), supports both deterministic and stochastic predictions.

## 2.2 Social & Scene-aware Modelling

**Approaches to Agent-Agent Interaction Modelling** The social dimension focuses on capturing agent-agent interactions, emphasizing how individuals or objects influence each other's movements within a shared space. Notably, some methodologies incorporate social pooling, concurrently with attention mechanisms Pang et al. (2021). Algorithms in this domain predominantly leverage various forms of Graph Neural Networks (GNNs) to encapsulate the social dynamics among agents. Some methods employ a fully connected undirected graph, encompassing all scene agents Xu et al. (2023); Gilles et al. (2021); Kosaraju et al. (2019). This approach, albeit comprehensive, escalates exponentially with the number of agents (nodes). Conversely, other methods opt for sparsely connected graphs, establishing connections solely among agents within a proximal range, thereby reducing the linkage count substantially Fang et al. (2021); Girase et al. (2021); Salzmann et al. (2020); Weng et al. (2021). In a similar vein, Yuan et al. (2021); Salzmann et al. (2020) proposes sparse, directional graphs, predicated on the premise that different agent types possess varying perceptual ranges. Regarding the optimal depth of GNN layers, Addanki et al. (2021) advocate for deeper graphs to enhance performance. This stands in contrast to the findings of Weng et al. (2021) and Liu et al. (2020), who posit that two layers are optimal. Nevertheless, this depth increases computational demands, particularly when agent nodes are numerous, posing challenges for autonomous vehicle applications reliant on edge devices for processing.

**Scene Embeddings** The scene dimension, extracted from the video frames, includes the low-level representation of the physical environment, obstacles, and any elements that could affect the agent's path, ensuring a comprehensive understanding of both social and environmental factors in predicting movement trajectories. Rasouli et al. (2021) and Mangalam et al. (2021) capture scene dimension with the help of semantic segmentation to delineate visual attributes of varied classes, subsequently elucidating their interrelations via attention. However, obtaining a panoptic segmentation mask, might not be always feasible. Also, this approach introduces a considerable dependency on the availability of additional segmentation maps, presenting a challenge in terms of data requirements and preprocessing efforts.

## 2.3 Temporal Dimension

Understanding the trajectory history of an agent significantly augments the predictive accuracy regarding its potential future path. Predominantly, ego-camera-based models are tailored to shorter temporal horizons and employ 3D Convolutional Neural Networks (3D-CNNs) Fang et al. (2021); Kotseruba et al. (2021). Some research, instead, adopts Hidden Markov Models (HMMs) for temporal analysis Cai et al. (2020). For scenarios necessitating extended time horizon considerations, more intricate structures are proposed, including Transformers Yuan et al. (2021); Xu et al. (2023); Giuliari et al. (2021b); Chen et al. (2021) and various forms of Recurrent Neural Networks (RNNs) Girase et al. (2021), including Long Short-Term Memory networks (LSTMs) Rasouli & Kotseruba (2023); Bhattacharyya et al. (2021); Fang et al. (2021) and Gated Recurrent Units (GRUs) Gilles et al. (2021). Both Transformer and RNN-based models have exhibited superior performance, often achieving state-of-the-art results in this domain. However, some of these models tend to address the temporal dimension in isolation from the social context. This segregated approach potentially results in information loss and contributes to an increased computational load, necessitated by the addition of separate components to handle the social dimension. Consequently, there emerges a pressing demand for integrated models capable of concurrently processing both temporal and social dimensions. A promising direction in this regard is the development of graph-aware transformers, which encapsulate the essence of both temporal dynamics and social interactions within a unified framework.

### 2.4 Graph-aware Transformers

Graph-aware transformers aim to compound the benefits of graphs (with their associated social embeddings) and of transformers, with their acclaimed attention mechanism and temporal embeddings. Notably, these advancements have predominantly catered to graph-centric datasets like ACTOR Tang et al. (2009) and CHAMELEON SQUIRREL Rozemberczki et al. (2021). Direct application of graph-aware transformers remains untouched in pedestrian trajectory forecasting, with prevalent methodologies leaning towards transformers processing embeddings emanating from graphs Li et al. (2022); Chen et al. (2023); Jia et al. (2023). There has been a discernible preference for using GNN and transformer blocks, rather than fully-integrated graph-aware transformers.

A comprehensive evaluation of numerous contemporary graph-transformer models across three graph-centric datasets is conducted in Müller et al. (2023). The analysis reveals a consistent pattern: models employing Random Walk for structural encoding exhibit superior performance across all tested datasets. Building on this empirical evidence, our approach utilizes Random Walk to encode the pedestrian graph, which is then seamlessly integrated into the transformer architecture. This integration is designed to yield a graph-aware transformer, thereby enhancing the model's capability to effectively capture and interpret complex pedestrian dynamics within various environments. To the best of the authors' knowledge, this is the first work towards utilising a graph-aware transformer to solve the trajectory prediction problem, opposing many methods which use graphs along with transformers.

## 3 Problem Formulation

The objective of trajectory prediction is to forecast a pedestrian's future position based on their observed historical trajectory. Formally, the historical trajectory of $A$ target agents is given as a sequence of coordinates:

$$\boldsymbol{X} = \{X_t^a \mid t \in (1, 2, \ldots, T_{obs}); \ a \in (1, 2, \ldots, A)\}, \tag{1}$$

where $X_t^a$ represents the position of agent $a$ at time $t$ over the past $T_{obs}$ time steps. For bird's-eye view (BEV) datasets, $X_t^a$ consists of 2D coordinates $\{x_t^a, y_t^a\}$. For egocentric view (EVV) datasets, it consists of bounding box coordinates $\{x_{1,t}^a, y_{1,t}^a, x_{2,t}^a, y_{2,t}^a\}$. Additionally, the corresponding visual input frames/images are given as $\boldsymbol{I} = \{I_t \mid t \in (1, 2, \ldots, T_{obs})\}$.

**Deterministic Setting**   The goal of ASTRA is to predict deterministic future trajectories for the pedestrians. In the deterministic setting, the model predicts future trajectory coordinates for each of the $A$ agents over the next $T_{pred}$ time steps:

$$\hat{\boldsymbol{Y}} = \{\hat{Y}_t^a \mid t \in (1, 2, \ldots, T_{pred}); \ a \in (1, 2, \ldots, A)\}, \tag{2}$$

where $\hat{Y}_t^a$ represents the predicted position of agent $a$ at future time $t$, and the ground truth trajectory is denoted as $\boldsymbol{Y} = \{Y_t^a \mid t \in (1, 2, \ldots, T_{pred}); \ a \in (1, 2, \ldots, A)\}$.

**Stochastic Setting**   In the stochastic setting, the goal is to learn a generative model parameterized by $\theta$ as $p_\theta(\mathcal{Y}|\boldsymbol{X}, \boldsymbol{I})$, which, given the historical observations $\boldsymbol{X}$ and $\boldsymbol{I}$, generates $K$ possible future trajectories:

$$\mathcal{Y} = \{\hat{\boldsymbol{Y}}^{(1)}, \hat{\boldsymbol{Y}}^{(2)}, \ldots, \hat{\boldsymbol{Y}}^{(K)}\}. \tag{3}$$

This enables the model to capture the inherent uncertainty in pedestrian motion by producing diverse trajectory hypotheses.

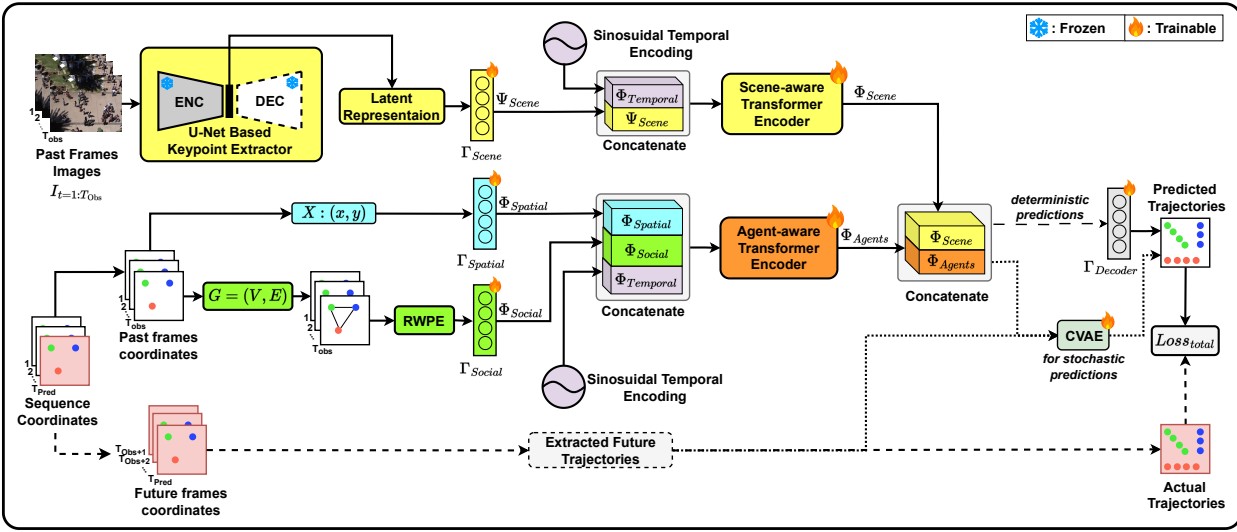

Figure 2: **Model Architecture.** Overview of ASTRA model architecture for pedestrian trajectory forecasting.

# 4 ASTRA Model

## 4.1 High-Level Overview

The encoder part of our model consists of two main components: A scene-aware component and an agent-aware component. While the former is dedicated to encoding the scene, and encapsulating the contextual details, the latter focuses on encoding the spatial, temporal, and social dimensions of the agents, as shown in Figure 2. The output from these two components is aggregated before being decoded to generate predicted future trajectories for deterministic or stochastic predictions.

## 4.2 Scene-aware Transformer Encoder

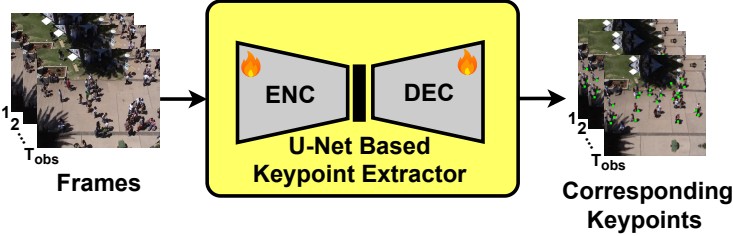

Figure 3: Pretraining U-Net based keypoint extractor.

In order to learn essential information about the scene's spatial layout and the positional dynamics of agents within it, the U-Net Ronneberger et al. (2015) is pre-trained to predict the location of pedestrians in the frame using the method detailed in Ribera et al. (2019), which utilizes a specialized loss function, Weighted Hausdorff Distance. This, in turn, helps the model learn a representation of the scene context (Figure 3), focusing on the pedestrians in the scene. This method of pre-training method allows the extraction of essential scene features involving pedestrians —since keypoints or locations of the pedestrians are always available in form of trajectories —without relying on explicit segmentation map annotations. This not only alleviates the need for labor-intensive annotation processes but also enables more efficient training on datasets where only pedestrian locations are provided.

A latent representation of pedestrian characteristics is obtained using a pre-trained U-Net encoder (Figure 2); this latent vector can include some crucial characteristics like spatial groupings and interactions with

the environment. The Grad-CAM visualizations (Figure 4) highlight this capability, showing that the pretrained model pays attention to regions with a high likelihood of pedestrian activity. This step is crucial as the U-Net extractor possesses the proficiency to discern both labelled and unlabelled pedestrians, depicted in green and red, respectively in Figure 4a.

More sophisticated schemes to generate the scene representation, like transformer-based architectures, and fusing social representations via gated cross-attention can also be considered but we leave exploring possibly more effective and sophisticated architecture designs as future work. The U-Net-based keypoint extractor is frozen after the pretraining (Figure 2) when used in ASTRA model architecture.

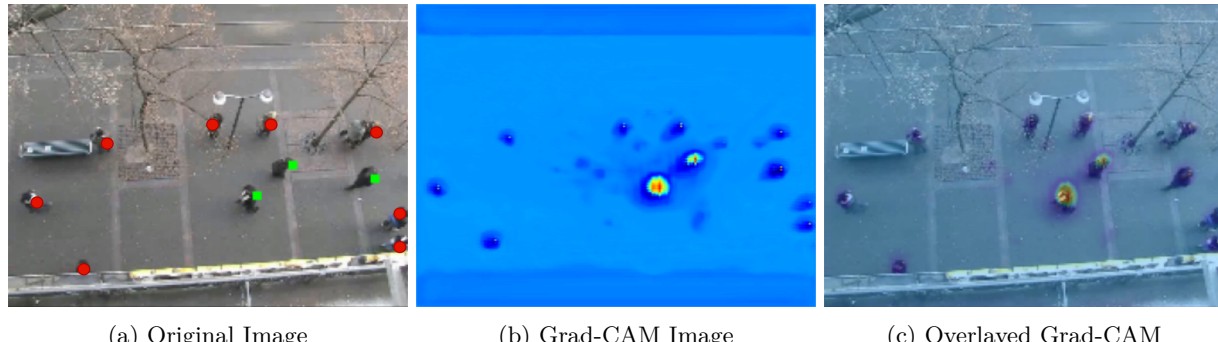

(a) Original Image          (b) Grad-CAM Image          (c) Overlayed Grad-CAM

Figure 4: **Grad-CAM visualizations**: In (a), the red circle indicates unlabelled pedestrians, while the green square highlights labelled pedestrians. In (b), the U-Net-based keypoint extractor focuses on unlabelled pedestrians as well, thereby capturing scene context from them too.

The latent representation of all frames ($\Psi_{\text{Scene}}$) is treated as input tokens to the scene-aware single-layer transformer encoder ($T_{\text{Scene-aware}}$), which in turn generates scene-aware embeddings ($\Phi_{\text{Scene}}$) for each frame. The single-layer transformer encoder architectural choice significantly contributes to the lightweight nature of our model. The resulting scene embedding is:

$$\Psi_{\text{Scene}} = \Gamma_{\text{Scene}} \left( \Upsilon_{\text{Encoder}}(\boldsymbol{I}) \right) \tag{4}$$

$$\Phi_{\text{Scene}} = T_{\text{Scene-aware}} \left( [\Psi_{\text{Scene}}; \Phi_{\text{Temporal}}] \right) \tag{5}$$

where the latent representation of past input frame images (I) projected using a Multi-Layer Perceptron (MLP) layer ($\Gamma_{\text{Scene}}$), and $\Upsilon_{\text{Encoder}}(.)$ denotes the encoder part of the U-Net.

The temporal encoding $\Phi_{\text{Temporal}}$, crucial for capturing the temporal dynamics within the observed frames, adopts the design of the traditional positional encoding Vaswani et al. (2017) and follows the work of Yuan et al. (2021).

### 4.3 Agent-aware Transformer Encoder

The second component is dedicated to encoding the different dimensions of each agent for all agents in the scene.

**Spatial Dimension**   The spatial coordinates **X** of each agent are linearly projected using an MLP layer ($\Gamma_{\text{Spatial}}$) to obtain the spatial encoding ($\Phi_{\text{Spatial}}$). The spatial encoding belongs to $\mathbb{R}^{A \times D_{\text{S}}}$, where $A$ is the number of agents and $D_{\text{Sp}}$ is the spatial feature dimension.

$$\Phi_{\text{Spatial}} = \Gamma_{\text{Spatial}}(\boldsymbol{X}) \in \mathbb{R}^{A \times D_{\text{Sp}}} \tag{6}$$

**Temporal Dimension**   Relying solely on spatial embeddings is insufficient, as agents occupying the same location in different frames would have identical spatial representations. To address this, we incorporate temporal encoding to distinguish agents across time.

To model temporal dependencies within pedestrian trajectories, we introduce temporal encodings for both agents and the scene. These encodings guide the network in capturing sequential information, similar to positional encodings in the Transformer architecture Vaswani et al. (2017). The temporal encoding $\Phi_{\text{Temporal}}$ belongs to $\mathbb{R}^{A \times D_{\text{T}}}$, where $A$ is the number of agents and $D_{\text{T}}$ is the temporal feature dimension.

$$\Phi_{\text{Temporal}}(t, i) = \begin{cases} \sin\left(\frac{t}{10000^{2i/D_{\text{T}}}}\right) & \text{if } i \text{ is even,} \\ \cos\left(\frac{t}{10000^{2i/D_{\text{T}}}}\right) & \text{if } i \text{ is odd.} \end{cases} \in \mathbb{R}^{A \times D_{\text{T}}} \tag{7}$$

**Social Dimension** Having the spatial and temporal dimensions of the agents is still not enough to understand their interaction in the scene. To capture the social dimension in this multi-agent environment, we generate a fully connected undirected graph between agents, in which the nodes are the agents' locations, and the edges between agents are the reciprocal of the distance. Consequently, the closer the agents are to each other, the stronger the link between them. Formally, we represent the social dimension using a graph $G = (V, E)$, where $V$ is the set of agents and $E$ is the collection of edges, with weights

$$e_{ij} = \frac{1}{d(v_i, v_j)} \tag{8}$$

where $d(v_i, v_j)$ is the distance between agents $v_i$ and $v_j$.

Subsequently, Random Walk Positional Encodings (RWPEs) Dwivedi et al. (2021) are used to capture the structural relationships between nodes in the graph, such as their proximity and connectivity patterns. RWPE leverages random walks to encode positional information, allowing the model to incorporate relational dependencies between agents in a data-driven manner. By considering the number of paths and transition probabilities between nodes, RWPE provides a meaningful representation of spatial interactions in a dynamic environment.

These RWPEs are then projected using a separate MLP ($\Gamma_{\text{Social}}$) to obtain social encodings ($\Phi_{\text{Social}}$). The social encoding belongs to $\mathbb{R}^{A \times D_{\text{So}}}$, where $A$ is the number of agents and $D_{\text{So}}$ is the social feature dimension:

$$\Phi_{\text{Social}} = \Gamma_{\text{Social}}\left(\text{RWPE}(\mathbf{G})\right) \in \mathbb{R}^{A \times D_{\text{So}}} \tag{9}$$

This preserves the graph structure of the agents while making the transformer encoder graph-aware. Furthermore, our method empowers the network to determine the significance of each agent relative to others autonomously.

While we do not claim to be the first to use transformers or graphs, we claim that we are the first to integrate RWPE directly into transformer tokens, creating a graph-aware transformer in the context of trajectory prediction.

**Aggregating** After computing the spatial, temporal, and social representations for each agent, our model concatenates them along the agent dimension. This ensures that each agent has a single feature vector consisting of a timestamp (temporal encoding), a social feature vector, and a spatial feature vector. The concatenated representation is then processed by an agent-aware single-layer transformer encoder ($T_{\text{Agent-aware}}$), which generates the final agent-aware embedding ($\Phi_{\text{Agents}}$).

$$\Phi_{\text{Agents}} = T_{\text{Agent-aware}}\left([\Phi_{\text{Spatial}}; \Phi_{\text{Temporal}}; \Phi_{\text{Social}}]\right) \in \mathbb{R}^{A \times (D_{\text{Sp}}+D_{\text{T}}+D_{\text{So}})} \tag{10}$$

### 4.4 Decoder

**Stochastic Decoding** To generate multiple stochastic trajectories, we learn a generative model, $p_\theta(\mathcal{Y}|\boldsymbol{X}, \boldsymbol{I})$ for which we adopted CVAE (Conditional Variational Auto Encoder). We train CVAE, similar to Yuan et al. (2021) and Yao et al. (2021), to learn the inherent distribution of future target trajectories

conditioned on observed past trajectories, by utilizing a latent variable $Z$. CVAE consists of three components - prior network ($p_\theta(Z|\tilde{X})$), recognition network (or posterior) ($q_\phi(Z|Y,X)$) and generation network ($g_\nu(\hat{Y}|Z)$), parameterized by $\theta$, $\phi$ and $\nu$ respectively. Here $\tilde{X}$ is the latent representation of X and I, obtained after concatenating $\Phi_{\text{Agents}}$ and $\Phi_{\text{Scene}}$ and $\hat{Y}$ is the output of the generation network and are the predicted future trajectories.

For CVAE, prior distribution ($p_\theta(z|\tilde{X})$) is parameterized by $\mathcal{N}(\mu_z^p, (\sigma_z^p)^2)$. The approximate posterior distribution ($q_\phi(z|Y,X)$) is parameterized by $\mathcal{N}(\mu_z^q, (\sigma_z^q)^2)$, where $\mu_z^p$ and $(\sigma_z^p)^2$ represent the mean and variance of the prior distribution and $\mu_z^q$ and $(\sigma_z^q)^2$ represent the mean and variance of the posterior distribution. During training, we sample latent variable ($z$) from the recognition network (posterior distribution) and feed it to the generation network ($(g_\nu(\hat{Y}|Z))$), whereas during testing we sample $z$ from the prior network (prior distribution). We use the reparameterization trick to sample $z$ through the mean and variance pairs of $(\mu_z^p, (\sigma_z^p)^2)$ and $(\mu_z^q, (\sigma_z^q)^2)$, respectively. KL divergence Loss ($\mathcal{L}_{\text{KL}}$) help in minimizing the difference between the distribution of latent variable(z) of prior and recognition network.

$K$ samples are drawn from the learned distribution and decoded using an MLP to obtain future trajectories. We optimize the parameters of the networks using the KL divergence which ensures that the prior network implicitly learns the dependency between future trajectories ($\boldsymbol{Y}$) and past trajectories ($\boldsymbol{X}$)

$$\mathcal{L}_{\text{KL}} = D_{KL}\big(q_\phi(Z \mid Y, X) \,\big\|\, p_\theta(Z \mid \tilde{X})\big) = D_{KL}\big(\mathcal{N}(\mu_{z_q}, (\sigma_z^q)^2) \,\big\|\, \mathcal{N}(\mu_{z_p}, (\sigma_{z_p})^2)\big). \tag{11}$$

**Deterministic Decoding**   For deterministic predictions, CVAE is skipped and the outputs of both the scene transformer ($\Phi_{\text{Scene}}$) and the agents' transformer ($\Phi_{\text{Agents}}$) are concatenated and directly passed through an MLP decoder ($\Gamma_{\text{Decoder}}$) to produce future trajectories ($\hat{Y}$) of the agents in the future frames as shown in Figure 2, namely:

$$\hat{\mathbf{Y}} = \Gamma_{\text{Decoder}}([\Phi_{\text{Scene}}; \Phi_{\text{Agents}}]) \tag{12}$$

## 4.5   Weighted Loss Function

We introduce a weighted-penalty strategy that can be applied to common loss functions used in trajectory prediction such as MSE and Smooth L1 Loss. The application of this strategy is through a dynamic penalty function $w(t)$, designed to escalate or de-escalate the significance of prediction errors as one moves further into the future. The definition of the weighted loss function is given by:

$$L_{\text{weighted}}(\hat{Y}, Y) = \sum_{t=1}^{T_{pred}} w(t) \cdot L(\hat{Y}_t, Y_t). \tag{13}$$

Where $\hat{Y}$ and $Y$ are the predicted and actual trajectories respectively, $T_{pred}$ denotes the number of prediction timesteps, $w(t)$ represents the dynamic weighting function at time $t$, and $L(\hat{Y}_t, Y_t)$ is the predefined loss function (e.g., MSE or SmoothL1 Loss) applied to the predicted and true positions at each time step $t$.

In time series data, as we move further into the future relative to the last observed data, the drift in predictions tends to increase, leading to higher errors. Motivated by this intuition, we initially penalized the predictions using linear and quadratic loss functions. These approaches showed improvements in overall prediction accuracy. However, upon closer analysis of the results, we observed that in some trajectories, there was a noticeable offset in the earlier parts of the predictions. To address this issue, we experimented with a parabolic weighting function for the penalty. Empirically, this approach outperformed the linear and quadratic strategies, yielding the most balanced and accurate predictions across the trajectories.

The weight function $w(t)$ (generally defined as $w(t) = f(t, T_{\text{pred}}, \alpha, \beta)$) is designed to be versatile, accommodating a spectrum of mathematical formulations that align with the specific needs of the predictive model. We used the parabolic weighted penalty, defined as:

$$w(t) = (\alpha - \beta) \cdot \left(2 \cdot \frac{t}{T_{pred}} - 1\right)^2 + \beta = 3\left(2 \cdot \frac{t}{T_{\text{pred}}} - 1\right)^2 + 1 \tag{14}$$

where $\alpha$ and $\beta$ are parameters that establish the bounds of the weighting function (during our experimentations and hyperparameter tuning, we found that $\alpha = 4$ and $\beta = 1$, as the optimal values), and $f$ is an adaptable function that governs the progression of weights at each timestep $t$.

In particular, the function $w(t)$ may be selected from various mathematical forms, such as linear, parabolic, or quadratic (discussed more in the supplementary materials). The choice of function enables the model to adjust the penalty progression in alignment with the anticipated prediction challenge at each timestep.

Hence, in the stochastic setting, the final loss consists of the weighted loss and the KL divergence loss from the CVAE. In contrast, in the deterministic setting, since the CVAE is not used, the KL term is omitted from the final loss:

$$\mathcal{L}_{\text{final}} = L_{\text{weighted}}(\hat{\boldsymbol{Y}}, \boldsymbol{Y}) + \mathcal{L}_{\text{KL}}. \tag{15}$$

## 5 Experiments

### 5.1 Datasets & Evaluation Protocols

For a comprehensive evaluation, we benchmarked our model on three trajectory prediction datasets; namely, ETH Pellegrini et al. (2009a), UCY Lerner et al. (2007), and PIE dataset Rasouli et al. (2019).

**ETH-UCY (Bird's Eye View)**  ETH and UCY offer a bird's-eye view of pedestrian dynamics in urban settings, including five datasets with 1,536 pedestrians across four scenes. For evaluation, we used their standard protocol; leave-one-out strategy, observing eight time steps (3.2s) and predicting the following 12 steps (4.8s).

**PIE (Ego-Vehicle View)**  In contrast, the PIE dataset provides an Ego-vehicle perspective, containing over 6 hours of ego-centric driving footage, along with bounding box annotations for traffic objects, action labels for pedestrians, and ego-vehicle sensor information Rasouli et al. (2019). A total of 1,842 pedestrian samples are considered with the following split: Training(50%), Validation(40%) and Testing(10%) Rasouli et al. (2019). Model performance is evaluated based on a shorter observational window of 0.5 seconds and a prediction window of 1 second, providing insights into the model's capability in rapidly evolving traffic scenarios Rasouli & Kotseruba (2023).

### 5.2 Evaluation Metrics

We used the standard evaluation metrics of ADE and FDE for ETH-UCY deterministic settings and minADE and minFDE for their stochastic settings. CADE, CFDE, ARB and FRB for PIE dataset. The supplementary material explains these metrics.

### 5.3 Hyperparameter Tuning & Hardware Settings

The hyperparameters were initially tuned on the ETH subset of the ETH-UCY dataset and subsequently applied across all subsets. The key architectural hyperparameters used in our model are as follows: spatial embedding dimension ($\Phi_{\text{Spatial}} \in \mathbb{R}^{16}$), U-Net scene latent representation ($\Psi_{\text{Scene}} \in \mathbb{R}^{16}$), temporal embedding dimension ($\Phi_{\text{Temporal}} \in \mathbb{R}^{8}$), and random walk embedding ($\Phi_{\text{Social}} \in \mathbb{R}^{8}$). The transformer encoder consists of a single layer with two attention heads and a dropout rate of 0.2. To keep the model lightweight, MobileNet v2 Sandler et al. (2018) encoder layers are used for U-Net to reduce computational overhead while maintaining feature extraction efficiency.

For training, we employ the AdamW optimizer with a weight decay of $5 \times 10^{-4}$ over 200 epochs. A cosine annealing scheduler is used, starting with an initial learning rate of $1 \times 10^{-3}$. All experiments were conducted on an NVIDIA DGX A100 system with 8 GPUs, each equipped with 80 GB of memory.

# 6 Results & Discussion

## 6.1 Quantitative Results

**ETH-UCY** For ETH-UCY, we compared our model results against several baselines. These comparisons are presented in Table 1 and Table 4, which contains results primarily sourced from the EqMotion (CVPR 2023) Xu et al. (2023) for deterministic predictions and LeapFrog (CVPR 2023) Mao et al. (2023) for stochastic predictions respectively. It is important to note that to provide a thorough comparative framework, we independently computed and included additional models Wang et al. (2022); Yao et al. (2021) to their respective tables as they were not originally part of the EqMotion or LeapFrog analysis. Our model advances the state-of-the-art on ETH-UCY, outperforming the EqMotion Xu et al. (2023) by improving predictive accuracy by approximately 27% on average for deterministic predictions as shown in Table 1 and approximately 10% on average improvement over LeapFrogMao et al. (2023) for stochastic predictions as shown in Table 4, highlighting the efficacy of our approach in diverse scenarios. While ASTRA maintains superior performance across most benchmarks, there are, however, some exception cases, like in the Hotel and Univ scenes, where a notable proportion of pedestrians remain largely stationary throughout both the observation and prediction windows, resulting in slightly inferior performance in these scenarios. We also highlight the effectiveness of utilizing frame encodings from U-Net, as demonstrated in Table 4.

**PIE** Similarly, we benchmarked our model against various established models for the PIE dataset. The comparative analysis is summarized in Table 2, with the reference results taken from PedFormer Rasouli & Kotseruba (2023), demonstrating an average improvement of 26%.

Table 1: **Deterministic Results**: ADE/FDE results for ETH-UCY baselines. Best in **bold**, second best underlined.

| Model | ETH | Hotel | Univ | Zara1 | Zara2 | Average |
|---|---|---|---|---|---|---|
| Linear | 1.33/2.94 | 0.39/0.72 | 0.82/1.59 | 0.62/1.21 | 0.77/1.48 | 0.79/1.59 |
| S-LSTMAlahi et al. (2016) | 1.09/2.35 | 0.79/1.76 | 0.67/1.40 | 0.47/1.00 | 0.56/1.17 | 0.72/1.54 |
| S-AttentionVemula et al. (2018) | 1.39/2.39 | 2.51/2.91 | 1.25/2.54 | 1.01/2.17 | 0.88/1.75 | 1.41/2.35 |
| SGAN-indGupta et al. (2018) | 1.13/2.21 | 1.01/2.18 | 0.60/1.28 | 0.42/0.91 | 0.52/1.11 | 0.74/1.54 |
| Traj++Salzmann et al. (2020) | 1.02/2.00 | 0.33/0.62 | 0.53/1.19 | 0.44/0.99 | 0.32/0.73 | 0.53/1.11 |
| TransFGiuliari et al. (2021b) | 1.03/2.10 | 0.36/0.71 | 0.53/1.32 | 0.44/1.00 | 0.34/0.76 | 0.54/1.17 |
| MemoNetXu et al. (2022b) | 1.00/2.08 | 0.35/0.67 | 0.55/1.19 | 0.46/1.00 | 0.37/0.82 | 0.55/1.15 |
| SGNetWang et al. (2022) | 0.81/1.60 | 0.41/0.87 | 0.58/1.24 | 0.37/0.79 | 0.31/0.68 | 0.56/1.04 |
| EqMotionXu et al. (2023) | 0.96/1.92 | 0.30/0.58 | **0.50**/1.10 | 0.39/0.86 | 0.30/0.68 | 0.49/1.03 |
| ASTRA (Ours) | **0.47/0.82** | **0.29/0.56** | 0.55/**1.00** | **0.34/0.71** | **0.24/0.41** | **0.38/0.70** |

Table 2: Results for PIE dataset.

| Model | CADE | CFDE | ARB | FRB |
|---|---|---|---|---|
| FOLYao et al. (2019) | 73.87 | 164.53 | 78.16 | 143.69 |
| FPLYagi et al. (2018) | 56.66 | 132.23 | - | - |
| B-LSTMBhattacharyya et al. (2018) | 27.09 | 66.74 | 37.41 | 75.87 |
| PIE$_{traj}$Rasouli et al. (2019) | 21.82 | 53.63 | 27.16 | 55.39 |
| PIE$_{full}$Rasouli et al. (2019) | 19.50 | 45.27 | 24.40 | 49.09 |
| BiPedRasouli et al. (2020) | 15.21 | 35.03 | 19.62 | 39.12 |
| PedFormerRasouli & Kotseruba (2023) | 13.08 | 30.35 | **15.27** | 32.79 |
| ASTRA(Ours) | **9.91** | **22.42** | 18.32 | **17.07** |

Table 3: **Ablation:** ADE/FDE for penalised vs. unpenalised loss functions on UNIV dataset using SOTA ASTRA's configuration

| Loss | Normal | Penalised |
|---|---|---|
| MSE | 0.58/1.13 | 0.57/1.00 |
| SmoothL1 | 0.57/1.15 | 0.55/1.00 |

## 6.2 Efficiency

Regarding the computational side, ASTRA has seven times fewer trainable parameters than the existing SOTA model LeapFrog Mao et al. (2023) as shown in Figure 1. It is important to note that the parameter count reported for ASTRA in Figure 1 includes the parameters from the U-Net, which is otherwise actually

Table 4: **Stochastic Results**: minADE$_{20}$ and minFDE$_{20}$ results for ETH-UCY baselines. Best in **bold**, second best underlined. NP- means unpenalised.

| Model | ETH | Hotel | Univ | Zara1 | Zara2 | Average |
|---|---|---|---|---|---|---|
| Social-GAN Gupta et al. (2018) | 0.87/1.62 | 0.67/1.37 | 0.76/1.52 | 0.35/0.68 | 0.42/0.84 | 0.61/1.21 |
| NMMP Hu et al. (2020) | 0.61/1.08 | 0.33/0.63 | 0.52/1.11 | 0.32/0.66 | 0.43/0.85 | 0.41/0.82 |
| STAR Yu et al. (2020a) | 0.36/0.65 | 0.17/0.36 | 0.31/0.62 | 0.29/0.52 | 0.22/0.46 | 0.26/0.53 |
| PECNet Mangalam et al. (2020) | 0.54/0.87 | 0.18/0.24 | 0.35/0.60 | 0.22/0.39 | 0.17/0.30 | 0.29/0.48 |
| Trajectron++ Salzmann et al. (2020) | 0.61/1.02 | 0.19/0.28 | 0.30/0.54 | 0.24/0.42 | 0.18/0.32 | 0.30/0.51 |
| BiTrap-NP Yao et al. (2021) | 0.55/0.95 | 0.17/0.28 | 0.25/0.47 | 0.22/0.44 | 0.16/0.33 | 0.27/0.49 |
| MemoNet Xu et al. (2022b) | 0.40/0.61 | **0.11/0.17** | 0.24/0.43 | 0.18/0.32 | 0.14/0.24 | 0.21/0.35 |
| GroupNet Xu et al. (2022a) | 0.40/0.76 | 0.12/0.18 | **0.22/0.41** | 0.17/0.31 | **0.12**/0.24 | 0.21/0.38 |
| SGNet Wang et al. (2022) | 0.47/0.77 | 0.20/0.38 | 0.33/0.62 | 0.18/0.32 | 0.15/0.28 | 0.27/0.47 |
| MID Gu et al. (2022) | 0.46/0.73 | 0.15/0.25 | 0.26/0.49 | 0.21/0.39 | 0.17/0.33 | 0.25/0.44 |
| Agentformer Yuan et al. (2021) | 0.45/0.75 | 0.14/0.22 | 0.25/0.45 | 0.18/0.30 | 0.14/0.24 | 0.23/0.39 |
| EqMotion Xu et al. (2023) | 0.40/0.61 | 0.12/0.18 | 0.23/0.43 | 0.18/0.32 | 0.13/0.23 | 0.21/0.35 |
| Leapfrog Mao et al. (2023) | 0.39/0.58 | **0.11/0.17** | 0.26/0.43 | 0.18/0.26 | 0.13/0.22 | 0.21/0.33 |
| TrajFine Wang et al. (2024) | 0.35/0.60 | **0.11**/0.18 | **0.22**/0.48 | **0.15**/0.30 | **0.12**/0.25 | **0.19**/0.36 |
| ASTRA(Non Penalised) | 0.37/0.49 | 0.24/0.34 | 0.37/0.52 | 0.23/0.32 | 0.16/0.23 | 0.27/0.38 |
| ASTRA(Without Image Input) | 0.29/0.39 | 0.18/ 0.29 | 0.29/ 0.43 | 0.17/0.26 | 0.14/0.2 | 0.21/ 0.31 |
| ASTRA(With Image Input) | **0.27/0.36** | 0.17/0.25 | 0.28/**0.41** | **0.15/0.23** | 0.13/**0.16** | 0.20/**0.28** |

frozen during the trajectory prediction phase. The complete ASTRA architecture, including all components, comprises 1.56M parameters, 652M FLOPs (floating-point operations), and 326M MACs (Multiply-Accumulate Operations). Even in configurations without U-Net pretraining, the deterministic variant has 13.52K parameters, 15.87K FLOPs, and 7.935K MACs, while the stochastic variant has 141.48K parameters, 165.416K FLOPs, and 82.708K MACs. This flexibility allows ASTRA to scale efficiently under strict resource constraints.

Additionally, we computed the model's running time, which takes 19.23ms to generate predictions for future trajectories. To further assess ASTRA's efficiency, we compare its running time, FLOPs, MACs, and number of parameters with three other models: AgentFormer, a transformer-based architecture similar to ours; Leapfrog, a state-of-the-art approach; and GroupNet, which has the fewest parameters among all compared models. As shown in the Table 5, ASTRA demonstrates strong efficiency compared to these alternatives.

Table 5: Model Parameters and Inference Time comparison. (**M** denotes Million and **ms** denotes Millisecond)

| Model | Parameters (M) | FLOPS (M) | MACs (M) | Inference Time (ms) |
|---|---|---|---|---|
| GroupNet | 3.14 | 806 | 403 | 105.78 |
| Agentformer | 6.78 | 3084 | 1542 | 521.34 |
| Leapfrog | 11.2 | 5695 | 2848 | 28.91 |
| **Astra (Ours)** | 1.56 | 652 | 326 | 19.23 |

## 6.3 Ablations

In the ablation study presented in Table 6, we evaluated the contribution of each component in our trajectory prediction model to ascertain their individual and collective impact on the performance metrics on the ETH-UCY (UNIV) dataset. Initially, the model incorporated only spatial information, which served as a baseline for subsequent enhancements. The sequential integration of temporal and social components yielded successive improvements, demonstrating their respective significance in capturing the dynamics of agent movement. The addition of the data augmentation technique (detailed in the supplementary material) further refined the model's performance, illustrating the value of varied training samples in enhancing generalization capabilities. Moreover, the incorporation of U-Net features contributed to a leap forward, highlighting the importance of context-aware embeddings in accurately forecasting agent trajectories. This progression

emphasizes the synergistic effect of combining heterogeneous data representations to capture the nuanced patterns of movement within a scene.

We also investigate the effect of using a Single Transformer Encoder versus Dual Transformer Encoders (Table 6, last two rows). For the Single Encoder setup, we merge $T_{\text{Scene-aware}}$ and $T_{\text{Agent-aware}}$ into a unified transformer encoder. While this configuration captures a joint representation of scene features and agent dynamics, it inherently limits the model performance, as can be inferred from Table 6.

Table 6: **Ablation:** ADE/FDE & $minADE_{20}/minFDE_{20}$ with variations in ASTRA's model components on UNIV dataset (where ✓: Component enabled, ×: Component disabled).

| Spatial | Temporal | Augmentation | Social | U-Net Features | Transformer Encoder | ADE/FDE | $minADE_{20}/$ $minFDE_{20}$ |
|---|---|---|---|---|---|---|---|
| ✓ | × | × | × | × | Single | 1.05/1.66 | 0.43/0.63 |
| ✓ | ✓ | × | × | × | Single | 0.86/1.47 | 0.39/0.54 |
| ✓ | ✓ | ✓ | × | × | Single | 0.67/1.17 | 0.31/0.48 |
| ✓ | ✓ | ✓ | ✓ | × | Single | 0.66/1.12 | 0.29/0.43 |
| ✓ | ✓ | ✓ | ✓ | ✓ | **Single** | 0.74/0.61 | 0.98/1.41 |
| ✓ | ✓ | ✓ | ✓ | ✓ | **Dual** | 0.55/1.00 | 0.28/0.41 |

The ablation study also extended to the evaluation of loss functions, comparing the effects of penalised versus unpenalised approaches. Penalized loss functions, designed to focus the model's attention on more critical prediction horizons, proved to be more effective in refining the predictive accuracy, as outlined in Table 3 and in Table 4 (ASTRA(NP)) for deterministic and stochastic setting respectively and the same can be observed in Figure 7.

Additionally, we compared the CLIP encoder —with added linear layer for fine-tuning in an end-to-end fashion —with our U-Net-based encoder, as shown in Table 7. Our model consistently performed on par with or better than CLIP while using significantly fewer parameters, MACs, and FLOPs, highlighting its efficiency. This can be attributed to the pre-training, which enables the model to focus specifically on pedestrians.

Table 7: Efficiency and quantitative ($minADE_{20}$ and $minFDE_{20}$) results for ETH-UCY baselines for different image encoders. (Million denotes **M**)

| Encoder | Parameter Count (M) | FLOPs (M) | MACs (M) | ETH | Hotel | Univ | Zara1 | Zara2 | Average |
|---|---|---|---|---|---|---|---|---|---|
| CLIP Radford et al. (2021) | 149.77 | 22540 | 11270 | 0.26/0.37 | 0.18/0.28 | 0.34/0.48 | 0.18/0.29 | 0.15/0.21 | 0.22/0.324 |
| **UNET (Ours)** | 1.56 | 652 | 326 | 0.27/0.36 | 0.17/0.25 | 0.28/0.41 | 0.15/0.23 | 0.13/0.16 | 0.2/0.282 |

## 6.4 Qualitative Analysis

We can clearly see the results of our prediction from Figure 5 for deterministic predictions and Figure 6 for stochastic prediction. Figure 7 exemplifies the proximity of our model's results to the ground truth, it also shows how using the weighted penalty strategy has yielded better results than the unpenalised one, highlighting the improved effectiveness of our strategy.

## 7 Conclusion & Future Work

We presented ASTRA, a model in the domain of pedestrian trajectory prediction, that outperforms the existing state-of-the-art models. This advancement renders ASTRA particularly suitable for deployment on devices with limited processing capabilities, thereby broadening the applicability of high-accuracy trajectory prediction technologies. ASTRA's adeptness in handling both BEV and EVV modalities further solidifies its applicability in diverse operational contexts. With the ability to produce deterministic and stochastic results, it enhances the predictive robustness and situational awareness of autonomous systems. Moving forward, we aim to extend the capabilities of the ASTRA model beyond pedestrian trajectory prediction to encompass

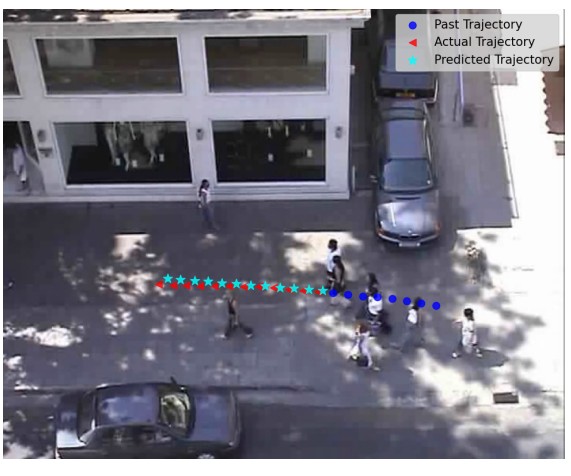

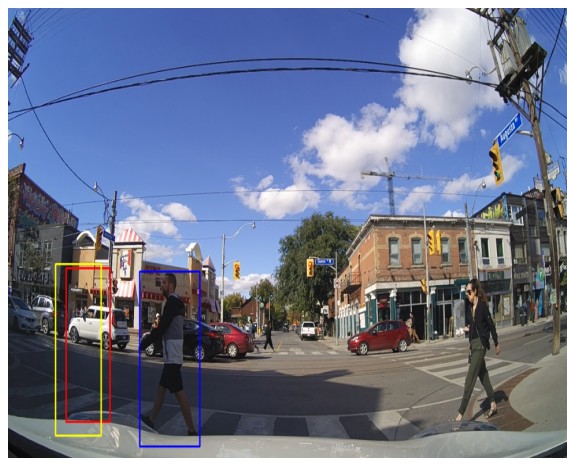

(a) BEV

(b) EVV

Figure 5: Sample images of the deterministic prediction from BEV datasets (a.) (ETH and UCY) and EVV dataset (b.) (PIE). The Red and Yellow bounding box indicates the ground-truth and predicted final position respectively and the Blue bounding box indicates the start position.

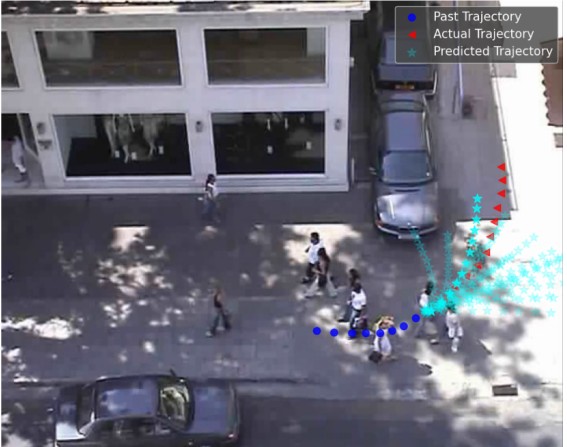

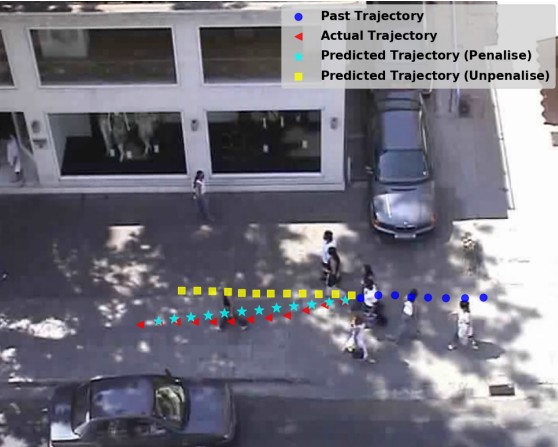

Figure 6: Sample images of the stochastic predictions from ETH-UCY Dataset.

Figure 7: Visual comparison of penalised and unpenalised loss on ETH-UCY, showing the enhanced performance of the former.

a broader range of non-human agents. This expansion will involve adapting the model to understand and predict the movements of various entities within shared environments using more sophisticated architectural design choices to encode the scene and its fusion with social dimension. By broadening our focus, we hope to contribute to the development of truly comprehensive and adaptive systems capable of navigating the complexities of real-world interactions among a wide array of agents.

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

# A Weighted Loss Function Details

## A.1 Loss Function Formulation

The loss for stochastic predictions is given in equations equation 16 and equation 17. $L_{\text{weighted}}(\mathcal{Y}_k, \boldsymbol{Y})$ is calculated similar to Equation 18.

$$L_{\text{weighted}}(\hat{\boldsymbol{Y}}, \boldsymbol{Y}) = \min_{k=1,\ldots,K} L_{\text{weighted}}(\mathcal{Y}_k, \boldsymbol{Y}) \tag{16}$$

$$\mathcal{L}_{\text{final}} = L_{\text{weighted}}(\hat{\boldsymbol{Y}}, \boldsymbol{Y}) + D_{KL}(\mathcal{N}(\mu_{z_q}, \sigma_{z_q}) \,||\, \mathcal{N}(\mu_{z_p}, \sigma_{z_p})) \tag{17}$$

For deterministic predictions, the final loss is the same as the weighted loss:

$$\mathcal{L}_{\text{final}} = L_{\text{weighted}}(\hat{Y}, Y) = \sum_{t=1}^{T_{pred}} w(t) \cdot L(\hat{Y}_t, Y_t), \tag{18}$$

where $w(t)$ represent the weighted penalty function (section A.2) and $L(\hat{Y}_t, Y_t)$ is the predefined loss function: MSE or Smooth L1 loss (discussed below).

**Mean square error (MSE)**

$$\text{MSE} = \frac{1}{N} \sum_{i=1}^{N} (y_i - \hat{y}_i)^2 \tag{19}$$

where $y_i$ and $\hat{y}_i$ represent, the actual and predicted coordinates, respectively. MSE penalises larger trajectory prediction errors more heavily, ensuring model accuracy in critical scenarios.

**Smooth L1 loss (SL1)**

$$\text{SL1}(y_i, \hat{y}_i) = \begin{cases} 0.5 \times (y_i - \hat{y}_i)^2 & \text{if } |y_i - \hat{y}_i| < 1 \\ |y_i - \hat{y}_i| - 0.5 & \text{otherwise.} \end{cases} \tag{20}$$

Unlike MSE, SL1 effectively balances the treatment of small and large errors. This loss is also less sensitive to outliers, due to its combination of L1 and L2 loss properties.

## A.2 Weighted Penalty Function

In many trajectory prediction tasks, errors at different time steps may have varying importance. For instance, predictions further into the future might be more uncertain, whereas errors in early predictions could be more critical for subsequent decisions. To address this, we introduce a weighted penalty function $w(t)$ that adjusts the loss function's sensitivity to prediction errors over time.

Our ablation focuses on three distinct penalty functions, $w(t, T_{\text{pred}}, \alpha, \beta)$: Linear, Quadratic, and Parabolic, which are parameterized by $\alpha$ and $\beta$, determined empirically as shown in Figure 8. Here, $t$ represents the current time step, and $T_{\text{pred}}$ denotes the total prediction horizon. Once the values of $\alpha$ and $\beta$ are determined, $w(t, T_{\text{pred}}, \alpha, \beta)$ can be simply denoted as $w(t)$.

Table 8 presents a quantitative analysis comparing the three penalty strategies as applied to the ETH-UCY (UNIV) dataset using SL1 loss. It can be observed that the Parabolic penalty yields better results compared to the other penalization strategies.

Figure 8 illustrates a comparison of the three weighted penalty strategies for a prediction window of 12 frames. The subsequent sections provide a detailed explanation of each of these penalty strategies.

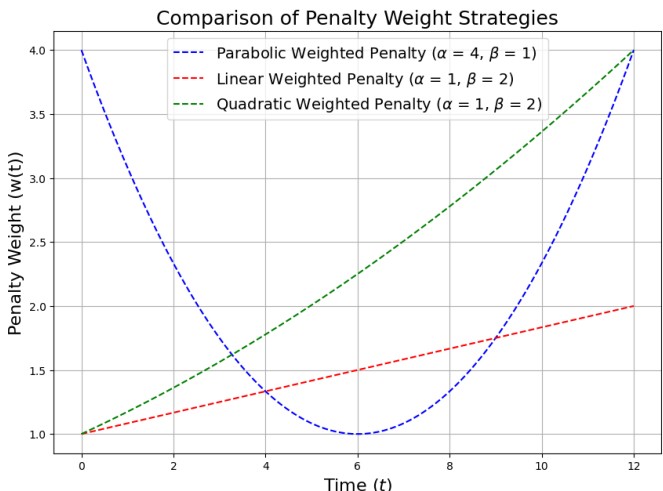

Figure 8: Comparison of various weighted penalty strategies

### A.2.1 Linear Weighted Penalty

The Linear Weighted Penalty employs a weight function, $w(t)$, that linearly increases from a start weight ($\alpha$) to an end weight ($\beta$), over the prediction period. This approach aims to progressively increase the penalty for prediction inaccuracies, particularly toward the latter part of the prediction horizon.

The weight function $w(t)$ is defined as:

$$w(t) = \alpha + \frac{t}{T_{pred}} \cdot (\beta - \alpha), \tag{21}$$

where $\alpha$ and $\beta$ are the weights assigned to the initial and final predicted time steps, respectively.

### A.2.2 Quadratic Weighted Penalty

The quadratic weighted penalty strategy intensifies the penalty in a quadratic manner as the difference between the prediction time and the past frames increases. This approach is more aggressive than the linear strategy, applying an exponentially increasing weight to errors in later prediction frames. The weight function $w(t)$ in this case is defined as:

$$w(t) = \left( \alpha + \frac{t}{T_{\text{pred}}} \cdot (\beta - \alpha) \right)^2 \tag{22}$$

### A.2.3 Parabolic Weighted Penalty

The Parabolic Weighted Penalty assigns the maximum weight, $\alpha$, to both the initial and final predicted time steps, highlighting their significance. Meanwhile, the minimum weight, $\beta$ ($\beta < \alpha$), is allocated to the midpoint of the prediction interval. This distribution forms a parabolic trajectory (shown in Figure 8) of weights across the prediction period, as defined by:

$$w(t) = (\alpha - \beta) \cdot \left( 2 \cdot \frac{t}{T_{pred}} - 1 \right)^2 + \beta, \tag{23}$$

### A.2.4 Augmentation

To enhance the robustness and generalization of our trajectory prediction model, we implement a data augmentation strategy inspired by Zamboni et al. (2022). This strategy applies random rotation and trans-

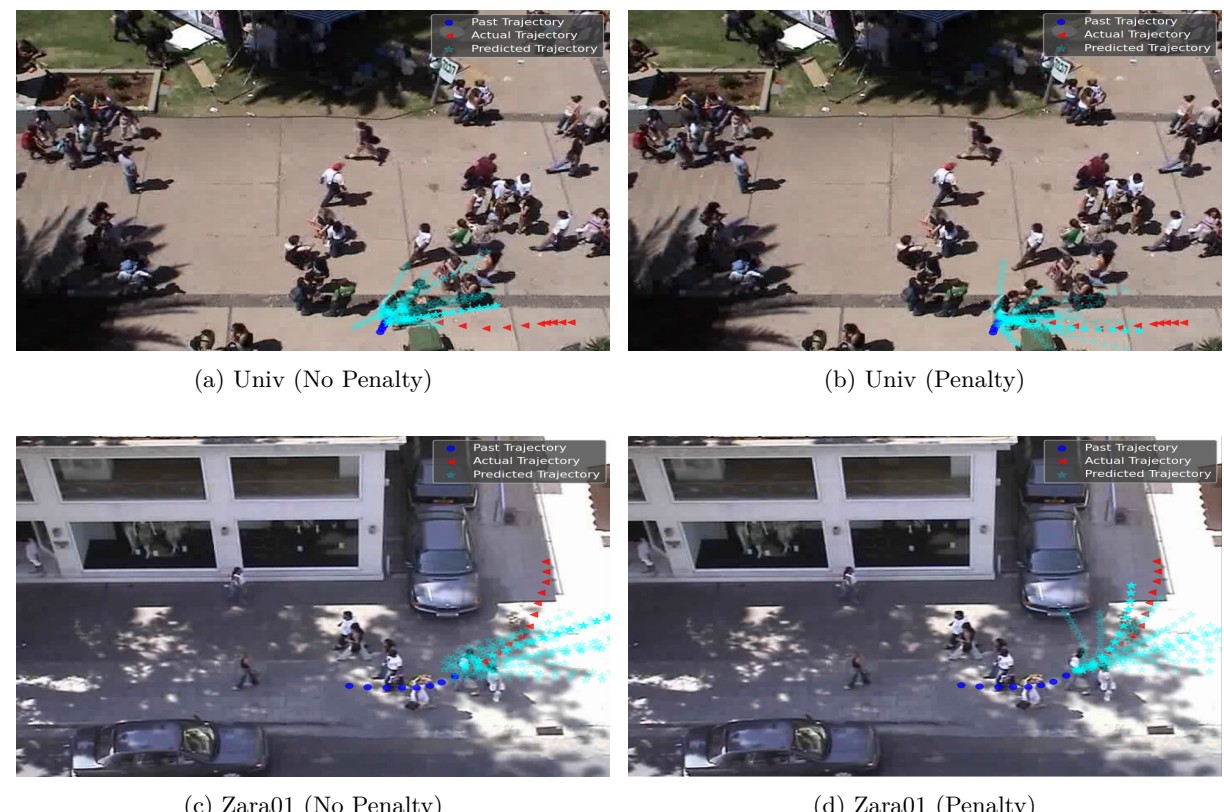

(a) Univ (No Penalty)                    (b) Univ (Penalty)

(c) Zara01 (No Penalty)                    (d) Zara01 (Penalty)

Figure 9: Qualitative comparison of unpenalised vs. penalised trajectories on ETH-UCY dataset in stochastic setting.

Table 8: **Ablation:** Comparing penalization strategies with SL1 loss on ETH-UCY (UNIV) dataset using ASTRA's SOTA configuration

| Loss | $\text{minADE}_{20}/\text{minFDE}_{20}$ |
|---|---|
| **Unpenalised** | 0.37/0.52 |
| **Linear** | 0.33/0.47 |
| **Quadratic** | 0.30/0.46 |
| **Parabolic** | **0.28/0.41** |

lation transformations to the trajectory sequences with a probability of 0.4, as illustrated in Fig. 11. By introducing these augmentations, the model becomes orientation-agnostic and better equipped to handle positional shifts in agents. The increased diversity in training data enables the model to learn more generalized representations of agent movements, improving its adaptability to real-world scenarios.

# B   Evaluation Metrics

## B.1   ETH-UCY

To evaluate our model on ETH-UCY, we used commonly employed evaluation metrics Xu et al. (2023); Mohamed et al. (2020); Chen et al. (2023); Yu et al. (2020b): ADE/FDE and $\text{minADE}_K/\text{minFDE}_K$. Average Displacement Error (ADE) computes the average Euclidean distance between the predicted trajectory and the true trajectory across all prediction time steps for each agent. $\text{minADE}_K$ refers to the minimum ADE out of K randomly generated trajectories and ground truth future trajectories. We also used the Final Displacement Error (FDE), which focuses on the prediction accuracy at the final time step. It computes the

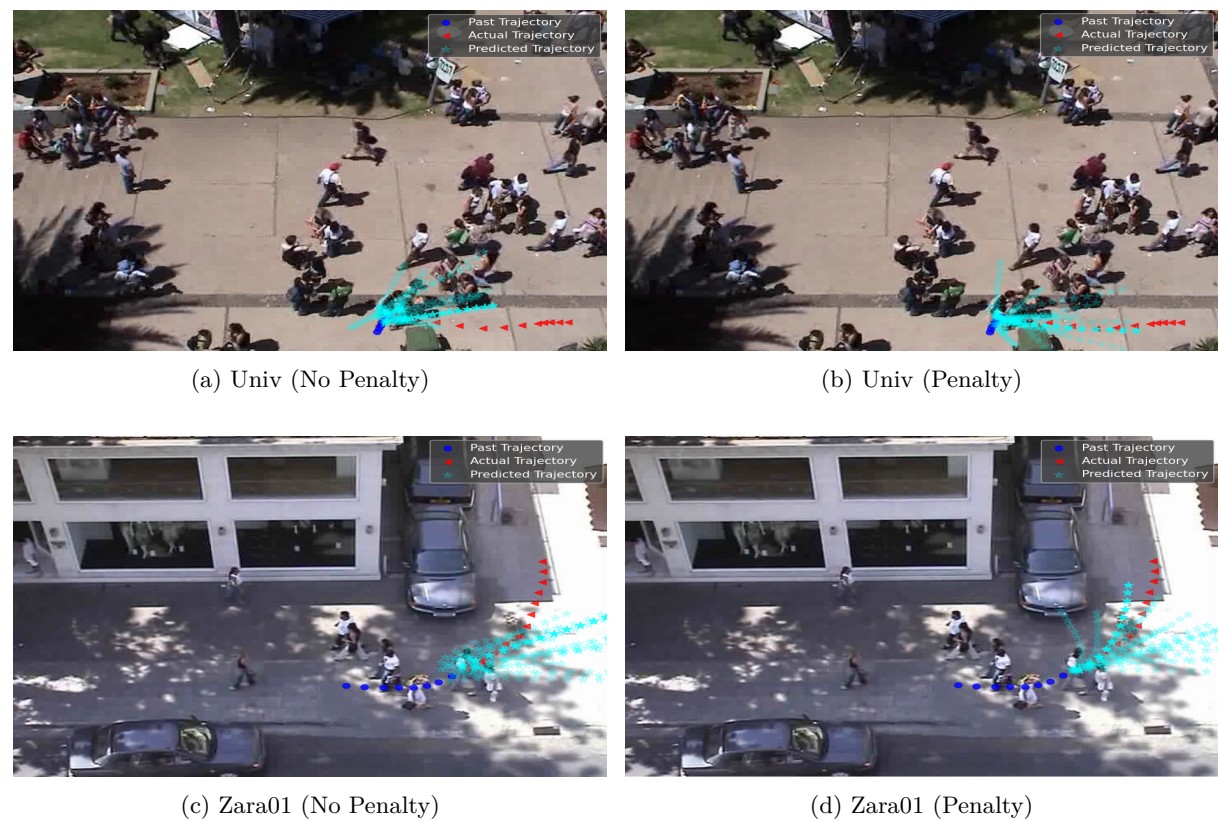

(a) Univ (No Penalty)                                    (b) Univ (Penalty)

(c) Zara01 (No Penalty)                                  (d) Zara01 (Penalty)

Figure 10: Qualitative comparison of unpenalised vs. penalised trajectories on ETH-UCY dataset in stochastic setting.

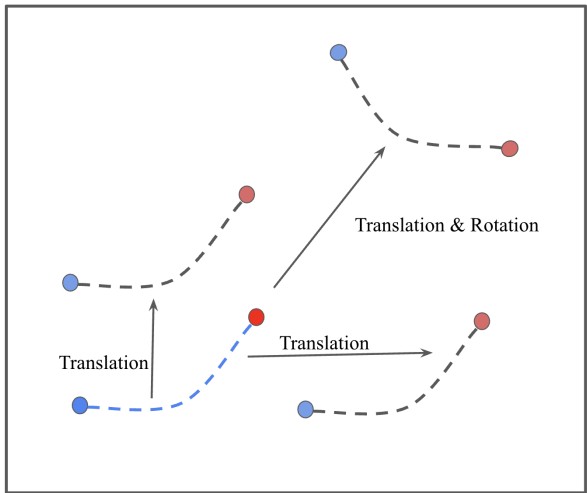

Figure 11: Illustration of data augmentation to trajectory sequences

Euclidean distance between the predicted and actual positions of each agent at the last prediction time step. $\text{minFDE}_K$ refers to the minimum FDE out of K randomly generated trajectories and ground truth future trajectories. For stochastic setting, $\text{minADE}_K$ and $\text{minFDE}_K$ metrics are used for evaluation.

$$\text{ADE} = \frac{1}{T_{pred}} \sum_{t=1}^{T_{pred}} \|Y_t^a - \hat{Y}_t^a\|_2. \tag{24}$$

$$\text{minADE}_K = \min_k \left( \frac{1}{T_{pred}} \sum_{t=1}^{T_{pred}} \|Y_t^a - \hat{Y}_{t,k}^a\|_2 \right) \tag{25}$$

$$\text{FDE} = \|Y_{T_{pred}}^a - \hat{Y}_{T_{pred}}^a\|_2 \tag{26}$$

$$\text{minFDE}_K = \min_k \left( \|Y_{T_{pred}}^a - \hat{Y}_{T_{pred},k}^a\|_2 \right) \tag{27}$$

### B.2 PIE

For the PIE Dataset, the ADE and FDE metrics are calculated based on the centroid of the bounding box Rasouli et al. (2019); Rasouli & Kotseruba (2023), denoted as Centre average displacement error for the bounding box (CADE) and Centre final displacement error for the bounding box (CFDE). In addition, we reported the average and final Root Mean Square Error (RMSE) of bounding box coordinates, denoted as ARB and FRB, respectively Rasouli et al. (2021).

## C Grad-CAM visualizations

Grad-CAM images were obtained by generating heatmaps overlaid onto the original image to aid in validating the relevance of highlighted regions. To obtain the Grad-CAM visualization, a single-channel output segmentation map was obtained from the pre-trained U-Net network, representing the probability of each pixel location being a keypoint Ribera et al. (2019). Probabilities were aggregated across all pixels, by comparing them with true keypoints and gradients of activation for the initial layer were extracted, similar to the approach taken by Vinogradova et al. (2020). Utilizing these gradients, a weighted average of the activation maps of the initial layer was computed to reconstruct the heatmap, similar to the method described in Selvaraju et al. (2017), for the Grad-CAM visualization. Overlaying this heatmap onto the original image highlights the regions that contribute significantly to the keypoint predictions made by the model.

## D Abbreviations and Mathematical Notation

To ease reading the paper, Table 9 and 10 list the abbreviations and the mathematical symbols mentioned in the paper, respectively.

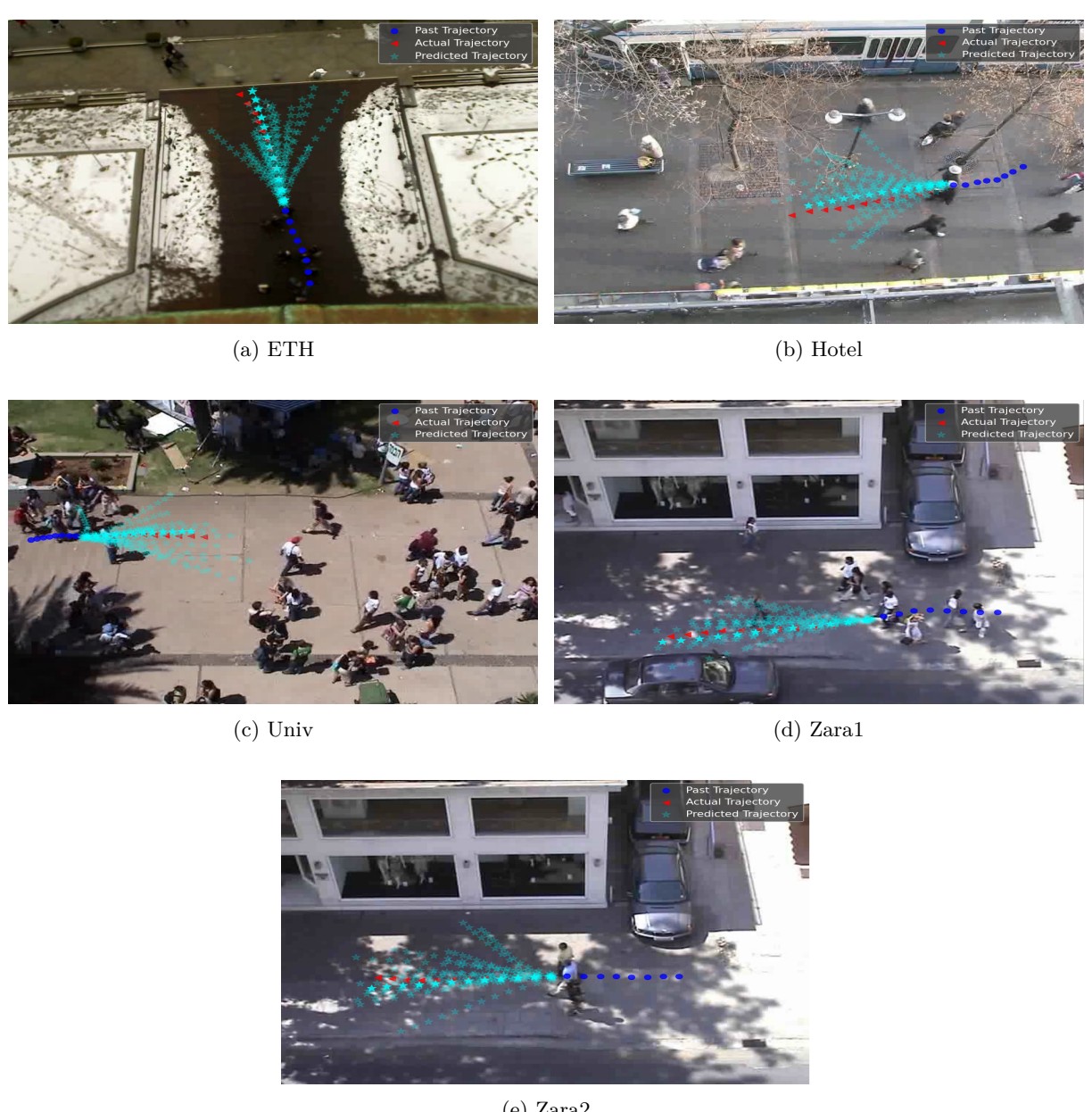

(a) ETH

(b) Hotel

(c) Univ

(d) Zara1

(e) Zara2

Figure 12: Trajectory visualizations for stochastic setting on ETH-UCY dataset (BEV)

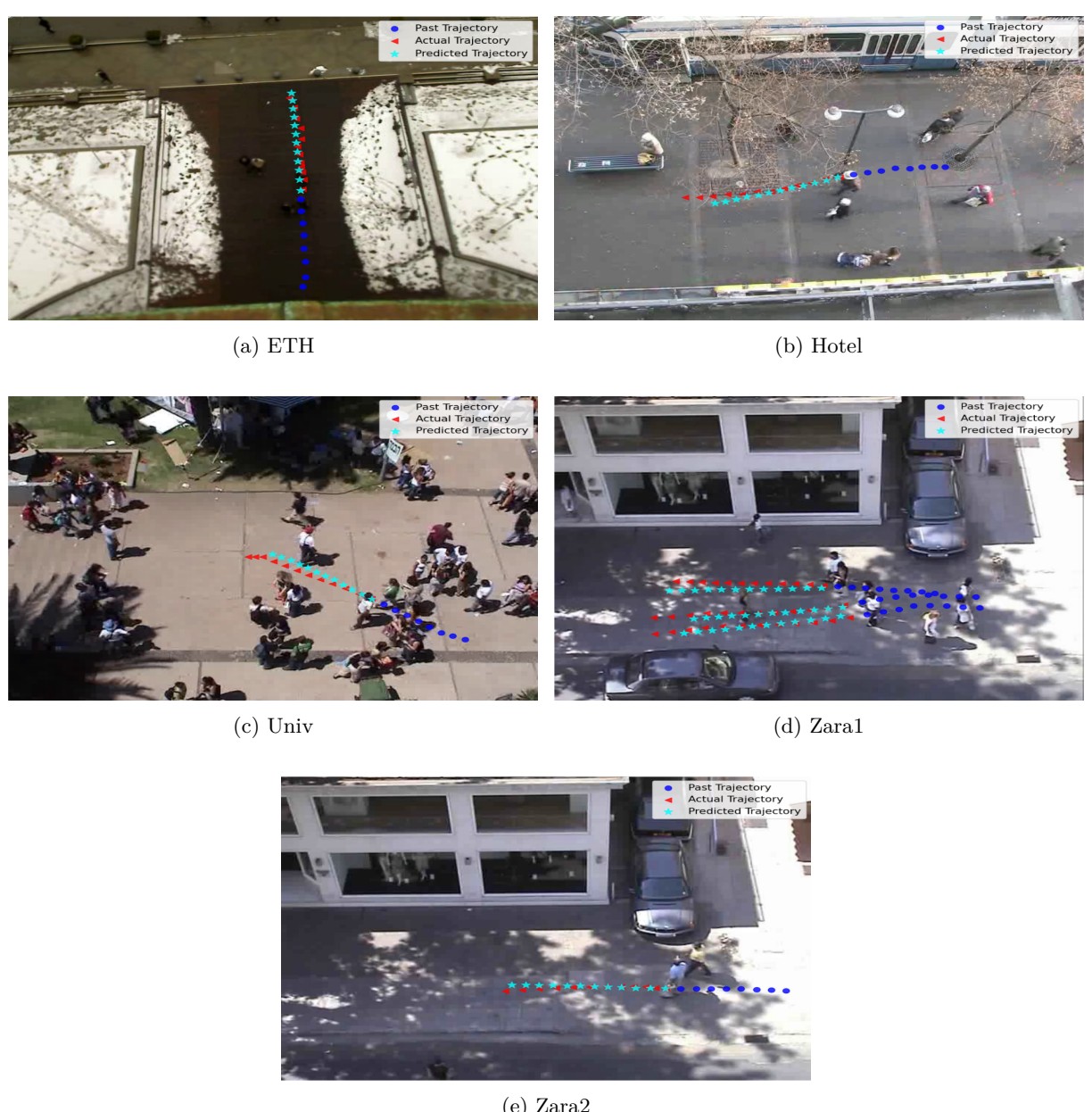

(a) ETH

(b) Hotel

(c) Univ

(d) Zara1

(e) Zara2

Figure 13: Deterministic trajectory visualizations on ETH-UCY dataset (BEV)

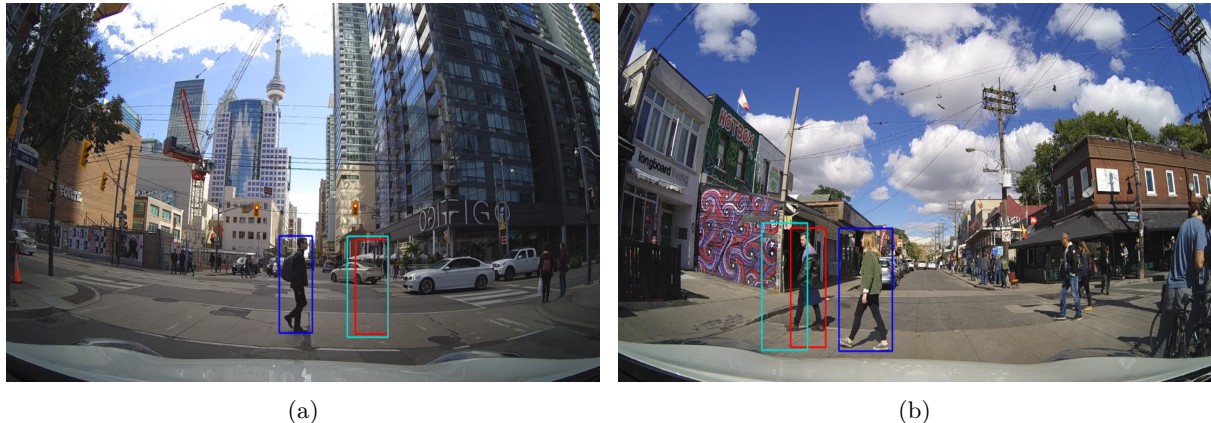

Figure 14: Trajectory Visualizations on PIE Dataset (EVV) where the red and cyan bounding box indicates the ground-truth and predicted final position respectively and the blue bounding box indicates the start position.

Table 9: Table of Abbreviations Used

| Abbreviation/Term | Description |
| --- | --- |
| ASTRA | Agent-Scene aware model for pedestrian trajectory forecasting |
| BEV | Bird's Eye View |
| EVV | Ego-Vehicle View |
| AV | Autonomous Vehicle |
| MLP | Multi-Layer Perceptron |
| CVAE | Conditional Variational Auto-Encoder |
| GNN | Graph Neural Network |
| RWPE | Random Walk Positional Encoding |
| MSE | Mean Square Error (Loss Function) |
| SL1 | Smooth L1 Loss (Loss Function) |
| ADE | Average Displacement Error |
| FDE | Final Displacement Error |
| CADE | Centre average displacement error for the bounding box |
| CFDE | Centre final displacement error for the bounding box |
| ARB | Average Root Mean Square Error for the bounding box |
| FRB | Final Root Mean Square Error for the bounding box |

Table 10: Table of Mathematical Symbols Used

| Symbols | Description |
|---|---|
| $N$ | Total number of predictions in MSE calculation |
| $\boldsymbol{X}$ | Observed trajectories of agents |
| $\boldsymbol{Y}$ | Groundtruth future trajectories of agents |
| $\boldsymbol{\hat{Y}}$ | Predicted trajectories of agents |
| $T_{\text{obs}}$ | Number of past time instants for observation |
| $T_{\text{pred}}$ | Number of future time instants for prediction |
| $\boldsymbol{I_{t=1:T_{\text{Obs}}}}$ | Sequence of past input frame images |
| $X_t^a$ | Observed coordinates for agent $a$ at time $t$ |
| $\hat{Y}_t^a$ | Predicted coordinates for agent $a$ at time $t$ |
| $A$ | Number of target agents |
| $e_{ij}$ | Edge weight in graph $G$ between nodes $i$ and $j$ |
| $d(v_i, v_j)$ | Distance between agents $v_i$ and $v_j$ |
| $w(t)$ | Weight function in weighted-penalty strategy |
| $w_{\text{start}}$ | Start weight in weighted-penalty strategy |
| $w_{\text{end}}$ | End weight in weighted-penalty strategy |
| $\boldsymbol{\Psi}_{\text{Scene}}$ | Latent representation of scene(past frame) obtained from U-Net encoder |
| $\boldsymbol{\Phi}_{\text{Scene}}$ | Scene-aware embeddings |
| $\boldsymbol{T}_{\text{Scene-aware}}$ | Scene-aware Transformer encoder |
| $\boldsymbol{\Upsilon}_{\text{Encoder}}$ | U-Net Encoder |
| $\boldsymbol{\Gamma}_{\text{Scene}}$ | Multi-layer Perceptron layer for Scene embeddings |
| $\boldsymbol{\Phi}_{\text{Temporal}}$ | Temporal encoding |
| $\boldsymbol{\Gamma}_{\text{Spatial}}$ | Multi-layer Perceptron layer for Spatial embeddings |
| $\boldsymbol{\Phi}_{\text{Spatial}}$ | Spatial embeddings |
| $\boldsymbol{\Gamma}_{\text{Social}}$ | Multi-layer Perceptron layer for Social embeddings |
| $\boldsymbol{\Phi}_{\text{Social}}$ | Social Embeddings |
| $\boldsymbol{T}_{\text{Agent-aware}}$ | Agent-aware Transformer encoder |
| $\boldsymbol{\Phi}_{\text{Agents}}$ | Agent-aware embeddings |
| $L_{\text{weighted}}(\hat{Y}, Y)$ | Weighted-penalty Loss Function |

