# OpenReview forum: "ASTRA: A Scene-aware Transformer-based Model for Trajectory Prediction"
_TMLR — Accepted by TMLR_

### Review · Reviewer_1y6m · 2024-10-10

**Summary Of Contributions:**

This paper introduces ASTRA, a trajectory forecasting model that can produce both deterministic and stochastic predictions, with seven times fewer parameters than the current state-of-the-art. The model leverages a U-Net-based feature extractor to generate scene-aware embeddings from input frames and employs a graph-aware transformer encoder to capture relationships between agents. Additionally, the authors propose a novel loss function that improves prediction accuracy by dynamically adjusting penalties based on prediction errors.

**Audience:**

Yes

**Broader Impact Concerns:**

No concern about the ethical implications of the work.

**Claims And Evidence:**

Yes

**Requested Changes:**

1. (Critical) Given the paper’s focus on building a more efficient model for trajectory forecasting, the authors should (i) provide a breakdown of the number of parameters for each component of the network, and (ii) report performance metrics like floating-point operations per second (FLOPS) or MAC (Multiply-ACCumulate) operations.
2. (Critical) The authors should also specify which dataset was used to tune the network’s hyperparameters and report these hyperparameters for easier reproducibility.
3. (Critical) The authors should consider including more recent trajectory forecasting models that were published before the paper’s submission (e.g., [1]).
4. (Would strengthen the work) The authors should expand the results section to explain why the model underperforms on certain datasets, such as Hotel and Univ.
5. (Would strengthen the work) Figure 3 includes a legend for frozen and trainable components, but the symbol is only used for the frozen components of the U-Net. The symbol for trainable components is missing, such as in the transformer encoders.

[1] Wang, Kuan-Lin, et al. "TrajFine: Predicted Trajectory Refinement for Pedestrian Trajectory Forecasting." Proceedings of the IEEE/CVF Conference on Computer Vision and Pattern Recognition. 2024.

**Strengths And Weaknesses:**

Strengths
1. The proposed method is evaluated on several datasets and compared against different baseline models (up to 2023), achieving the best performance on most datasets.
2. Compared to state-of-the-art methods, ASTRA’s reduced number of parameters represents a step toward making these models applicable to real-world applications.

Weaknesses
1. The results section would be improved by providing a more detailed explanation of why ASTRA outperforms state-of-the-art methods on most datasets but falls short on the Hotel, Univ, and Zara2 datasets.
2. The paper does not clearly explain how the network's hyperparameters were selected.
3. The model is claimed to be suitable for deployment on devices with limited processing power, however, the paper lacks a table showing performance metrics like floating-point operations per second (FLOPS) to support this claim.

---

> ### Author Response · Authors · 2025-02-14
>
> * Thank you for the comments, we added the FLOPs, inference time, and parameter count for our model and number of SOTA models.
>
> | Model         | Parameters (Millions) | FLOPS (Millions) | MACs (Millions) | Inference Time (ms) |
> |---------------|-----------------------|------------------|-----------------|----------------------|
> | **GroupNet**      | 3.14                  | 806              | 403             | 105.78               |
> | **Agentformer**   | 6.78                  | 3084             | 1542            | 521.34               |
> | **Leapfrog**      | 11.2                  | 5695             | 2848            | 28.91                |
> | **Astra (Ours)**  | 1.56                  | 652              | 326             | 19.23                |
>
>
>
> * We tuned the hyperparameters on the ETH subset and then applied them on the other datasets and subsets, we have added the main hyperparameters to the text.
> * We added: TrajFine to the comparison table, explanation of underperformance on certain dataset, and a sign to distinguish trainable components on Figure 2.

---

### Review · Reviewer_muft · 2024-10-29

**Summary Of Contributions:**

This paper proposes a neural network architecture, ASTRA, for forecasting the future trajectories of pedestrians from a bird's eye or vehicle perspective. ASTRA combines information from two sources of data, past pedestrian trajectories (X) and past images (I), to improve prediction performance. The model extracts features from these two sources of data (X and I) through two different neural networks: (1) U-Net + Transformer and a (2) Graph Neural Network + Transformer. These features are concatenated and passed through a decoder which predicts future trajectories. ASTRA can output a single prediction or multiple predictions through the use of a conditional variational auto-encoder (CVAE).

The paper claims the following contributions:
1) The two feature extraction heads are able to encode two different data sources to improve overall performance. The author's claim that this is an advantage where one source is unavailable.
2) ASTRA is trained with a weighted trajectory loss function that improves performance.
3) ASTRA contains 7 times fewer parameters than prior models which is more suitable for on-device processing.

**Audience:**

Yes

**Claims And Evidence:**

No

**Requested Changes:**

1. Experiments showing inference speed and memory footprint on a resource-limited device
2. Experiments showing performance without either data source (images or trajectories)
3. Address the definition of stochastic and multi-modal prediction as per above section.
4. Improve description of the weighted trajectory loss function in main paper.
5. Correct the description of KL divergence term and its role in the CVAE. Provide details on how the multi-modal sampling occurs through the CVAE.

**Strengths And Weaknesses:**

**Strengths**:
1. **Novelty**: The paper introduces a new architecture that incorporates multiple data sources. This is an interesting approach that attempts to resolve some practical challenges, such as data limitations in certain real world scenarios.
2. **Empirical results**: The experiment section suggest that this model outperforms almost all baseline models across a wide range of datasets.


**Weaknesses**:
1. **Claims of suitability for on-device processing**: The paper emphasizes the "number of parameters" as a key reason for being more suitable for on-device processing. However, parameter count is a major consideration from a training perspective, rather than a deployment perspective. To address this claim, the paper should include experiments and discussion on *inference speed* on a resource-limited device and *memory footprint* of ASTRA in comparison to other models. These are more important from a deployment perspective.
2. **Claims of usefulness during data absence**: The authors claim that prior works are problematic where data is unavailable, such as BEV. The experiment section does not provide evidence that ASTRA can perform in the absence of either data source.
3. **Stochasticity and multi-modal prediction**:  In Sec 2.1, the definition of stochastic prediction should not be equivalent to multi-modal prediction. Multi-modal prediction refers to predicting multiple modes or distributions (e.g. Gaussians) and can be used without the addition of randomness or generative models. Take for instance, the Gaussian Mixture Model (GMM). This section is also missing related lines of work on mixture density networks and multiple choice learning that are used in trajectory prediction.  See [1] as an example.
4. **Description of weighted trajectory loss function**: The description of the weighting function in Equation 10 on page 8 is extremely vague and could encapsulate almost anything. I do not think this description is sufficiently concise for something that is claimed to be a key contribution in the introduction. The exact functions are actually in the appendix, however the discussion is poor and mostly empirical.
5. **Description of KL Divergence for CVAE**:  There is a misunderstanding of the KL divergence term in Equation 11. It seems  that the KL divergence term arises due to the use of a CVAE, as per Figure 3 and Appendix A. There is a sentence that implies that the KL divergence helps the model learn the "dependency between Y and X". This is not true. The KL divergence term regularizes the latent distribution by encouraging the learned latent distribution to approximation a prior distribution (usually a Gaussian). The KL divergence term does not enable the generation of multiple plausible trajectories as suggested. Also the "contrasting" of $\mu_{z_q}$ and $\mu_{z_p}$ does not guide the model towards accurate predictions either. These descriptions mischaracterized the role of the KL divergence and should be removed.

**Reference**:
1. Makansi, Osama, et al. "Overcoming limitations of mixture density networks: A sampling and fitting framework for multimodal future prediction." Proceedings of the IEEE/CVF Conference on Computer Vision and Pattern Recognition. 2019.

---

> ### Author Response · Authors · 2025-02-14
>
> * Thank you for the comments, we added the FLOPs, inference time, and parameter count for our model and number of SOTA models.
>
> | Model         | Parameters (Millions) | FLOPS (Millions) | MACs (Millions) | Inference Time (ms) |
> |---------------|-----------------------|------------------|-----------------|----------------------|
> | **GroupNet**      | 3.14                  | 806              | 403             | 105.78               |
> | **Agentformer**   | 6.78                  | 3084             | 1542            | 521.34               |
> | **Leapfrog**      | 11.2                  | 5695             | 2848            | 28.91                |
> | **Astra (Ours)**  | 1.56                  | 652              | 326             | 19.23                |
>
> * Experiments without images are done and results are shown in Table 2 (ASTRA(Without Image Input)), however experiments without trajectories are less necessary because the it is expected that the performance will significantly drop without the trajectory input (Table 2 shows that the trajectory has a higher contribution to the performance than the images).
> * We defined the stochastic and deterministic prediction, and improved the description of weighted loss function as well as the KL term.

---

### Review · Reviewer_ytj4 · 2025-01-31

**Summary Of Contributions:**

This paper introduces a “lightweight” model for trajectory prediction. At each timestep, trajectories are represented by matrices of shape $A \times 2$ (coordinates) or $A \times 4$ (bounding boxes) where A is the number of agents (pedestrians). In addition to spatial embeddings, of these trajectories, The model incorporates

* Scene embeddings from a pretrained UNeT trained on keypoint detection that encodes the raw pixels.
* Social embeddings using an agent-to-agent graph based representation.
* Ttemporal position embeddings” in addition to “Spatial embeddings”.

A conditional VAE is also trained to output stochastic trajectories. Further, the authors introduce a timestep dependent weighting function in the loss which can upweight/downweight the loss in the future.

Quantitative results are reported on the ETH-UCY benchmark where the model outperforms all other baselines with lower parameter count on the average score.

**Audience:**

Yes

**Broader Impact Concerns:**

noo

**Claims And Evidence:**

Yes

**Requested Changes:**

Clarity
-----

* Please use notation such as $\phi \in R^{T \times D}$ to denote various embeddings such as the spatial, social and temporal in Section 3. This would make it clearer if these are global (1-D) or include spatial information (2-D).
* Along what axis does the self-attention in the transformer encoder operate? Is it the temporal dimension? Again, clarifying the dimensions of different vectors will be super helpful?
* How are the $T_{obs}$ dimensions to the inputs of the transfomer folded into a single output vector?
* Augmentation seems very important in Table 3 but is mentioned in Sec 4.4.3. What sort of augmentation is this and do other works use this as well?
* How are the hyperparameters  tuned?
* What is the precise $f$ used in Section 3.1.7? How are alpha and beta selected? Is it tuned for every dataset? Please highlight the main $f$ used in the paper.
* Eq 6): What are the dimensions of the graph representation G and the output of RWPE(G)?
* A high level description of RWPE would be helpful.
* Please describe the architecture of the CVAE in the main paper. How are the posterior, prior and decoder parametrized? What are the inputs to these different networks and what architectures do they follow?
* Page 3: Alleviates the data requirements and preprocessing efforts highlighted earlier. It’s unclear what this means and why using a keypoint model instead of a segmentation model helps to do this?
* Page 3: incorporating a penalty component is described here, but what this actually means is described as late as Section 3.1.7.
* This is in contrast to Yuan et al. (2021) which does not build a graph and does not preserve the social structure; they distinguish between self-agent and all other agents, then they treat all other agents the same without encoding the positional or structural encodings. Does this mean Yuan et. al (2021) do not have a graph representation for the input coordinates?
* For ablations in Table 3, the corresponding features are removed but the transformer Encoders are retrained. Is that correct?
* How was the keypoint model pretrained?
* It seems two separate models are needed for deterministic and stochastic predictions?

Novelty
--------

I understand that the loss weighting and keypoint latent representations are contributions of this paper? It’s a bit unclear what the other contributions are and which were based on prior work. Some concrete examples include the usage of graph representations to output $\phi_{social}$, usage of CVAE to model stochastic trajectories, sinusoidal temporal embeddings. The paper can state this upfront and provide citations.


Architecture
--------

* Are two transformer encoders (Scene-aware and Agent-aware) necessary with separate parameters? What happens if you concatenate all features and feed them to a single transformer? This ablation has to be compute adjusted so once has to train a bigger single transformer to make a fair comparison
* Have the authors considered using the keypoints itself rather than the latent representation?

**Strengths And Weaknesses:**

### Strengths

* The model incorporates scene embeddings from a pretrained keypoint detector which is interesting.
* The proposed architecture appears simple which is a positive.
* The reported results are quite strong.
* The same architecture + training seems to transfer from the ETH dataset (Birds Eye View) to the PIE Dataset (Ego Vehicle)

### Weaknesses

* The clarity of the paper can be significantly improved.
* After reading the architecture, it's unclear what elements are based on prior work and what are the authors' novel contributions.
* Some important details of the paper are relegated to the supplementary or not clearly explained.

I request the authors to improve the paper based on my suggestions below

---

> ### Author Response · Authors · 2025-02-14
>
> * Thanks for the comments, we added the dimensions to the vectors’ notation for a better understanding.
> * The self attention transformer operates along the agents’ axis, so that when the temporal, social, and spatial vectors are concatenated, it will output agents vectors, each vector containing a temporal part, social part and spatial part of a certain agent in the scene.
> * We added text and figure to better explain the augmentation part in the supplementary material.
> * Similar to other hyperparameters, Alpha and Beta were selected empirically by tuning on ETH subset,
> * We used the vanilla CVAE without changing its structure, that is why we did not mention it, however, we added it now to the main paper.
> * For ablations in Table 3, the corresponding features are removed but the transformer Encoders are retrained. Is that correct? Yes
> * We added a description of how the keypoint model was pretrained.
> * Yes, two separate models are needed for stochastic and deterministic, one with CVAE block and one without. This is to have a fair (one to one) comparison with the deterministic SOTA models which does not model the stochastic dimension and does not have a stochastic block. However, outside the experiments, our stochastic model can be used for stochastic and deterministic (by taking the most probable trajectory) results.
> * We adapted the temporal modelling from Agentformer, however, to the best of our knowledge, the idea of embedding the graph representation in the transformer to have a graph-aware transformer was new in the trajectory prediction domain.
> * We followed your suggestion of comparing the current model (two transformers) against having one transformer in Table 3 (Transformer Encoder).
> *We fixed the writing and clarity comments like the dimensions of the vectors
> * Does this mean Yuan et. al (2021) do not have a graph representation for the input coordinates? Yes, AgentFormer does not explicitly construct a graph representation for the input coordinates like some other trajectory prediction models that rely on graph-based relational reasoning (e.g., Social-GAN, Trajectron++, MemoNet). Instead, AgentFormer models the interactions between agents implicitly using a transformer-based architecture.

---

### Review · Reviewer_Vspv · 2025-01-31

**Summary Of Contributions:**

This paper introduces Astra, a new model that focuses on human trajectory forecasting. The main contribution are a novel model architecture dedicated to human trajectory prediction and a penalization scheme for the training loss function.

The author applies their Astra model to the ETH, UCY, and PIE datasets. Their empirical evaluation, both in deterministic and stochastic settings, shows the advantage over previous work on the proposed approach.

**Audience:**

Yes

**Broader Impact Concerns:**

No concen

**Claims And Evidence:**

No

**Requested Changes:**

- Ablation to validate the different design choices of the model, graph-aware transformer, compare different scene embedders…
- Add new results to better show the advantage of Astra in terms of training/inference runtime.
- Improve the clarity of the paper (see weaknesses).

**Strengths And Weaknesses:**

*Strength:*
- The paper shows strong performance compared to previous works on three human trajectory datasets.

*Weaknesses:*
- The main weakness of the paper, in my opinion, is that it’s not clear what the main factor contributing to the performance is. Despite the ablation in section 4.4, some of the technical decisions are not tested, i.e., for instance, what is the effect of using a ‘graph-aware’ transformer instead of using a streamlined transformer architecture?

- Some of the claims could be better supported:
    - Authors claim that their model is lightweight but only focus on parameter count. What is the advantage of Astra in terms of training and testing FLOPs?
    - Authors hypothesize that “transformers, by their inherent design, may pose potential challenges in preserving information, as they do not inherently accommodate the graph structure in their input,” but don’t verify it empirically.
    - Authors propose a scene-aware embedding using a UNet as a contribution but don’t compare it with other ways to extract scene information. One could potentially use DINOv2 or CLIP features to encode visual information
    -  It would be nice to validate the graph-aware architecture empirically, compared to a streamlined transformer architecture.

- The clarity of the paper could also be improved:
   - What is the main contribution of the paper? Is it a novel way to combine existing architecture components, or is it a specific architecture component like the graph-aware transformer or the UNet scene encoder?
   - The model description could use more details. For instance, it’s not clear how you extract a fixed-length representation from the transformer scene and agent representation.
   - While the author propose loss penalty as a contribution, there is no description of this part in the main text, only in appendix.

---

> ### Author Response · Authors · 2025-02-14
>
> * Thanks for the comments, we have added a comparison table (Table 7) depicting our model (UNet-based embedding) with the CLIP-based embeddings
>
> Efficiency and Quantitative (minADE₁₀ and minFDE₁₀) Results for ETH-UCY Baselines for Different Image Encoders
> (Million denotes **M**)
>
> | **Encoder**       | **Parameter Count (M)** | **FLOPs (M)** | **MACs (M)** | **ETH**  | **Hotel** | **Univ**  | **Zara1** | **Zara2** | **Average**  |
> |-------------------|------------------------|--------------|-------------|----------|----------|----------|----------|----------|------------|
> | CLIP         | 149.77                  | 22540        | 11270       | 0.26/0.37 | 0.18/0.28 | 0.34/0.48 | 0.18/0.29 | 0.15/0.21 | 0.22/0.324  |
> | **UNET (Ours)**  | 1.56                     | 652          | 326         | 0.27/0.36 | 0.17/0.25 | 0.28/0.41 | 0.15/0.23 | 0.13/0.16 | 0.2/0.282   |
>
>
> * To validate the effectiveness of the graph part, we reported the results with and without the graph part in Table 5.
> * The paper was restructured and we made sure the same topic is not mentioned multiple times at different sections through the paper.
> * The main contribution is injecting the graph representation into the transformer to have a graph-aware transformer, introducing a weighted loss function to improve the final predictions, and the optimised model which outperformed SOTA models while being lighter.
> * We added the FLOPs, inference time, and parameter count for our model and number of SOTA models.
> | Model         | Parameters (Millions) | FLOPS (Millions) | MACs (Millions) | Inference Time (ms) |
> |---------------|-----------------------|------------------|-----------------|----------------------|
> | **GroupNet**      | 3.14                  | 806              | 403             | 105.78               |
> | **Agentformer**   | 6.78                  | 3084             | 1542            | 521.34               |
> | **Leapfrog**      | 11.2                  | 5695             | 2848            | 28.91                |
> | **Astra (Ours)**  | 1.56                  | 652              | 326             | 19.23                |

---

### Author Response · Authors · 2025-03-30
**Regarding further decision**

Dear chairs,

Could you pl let us know regarding the tentative schedule of further decision of this paper. It is already quite a few months since we submitted the paper (submutted in september 2024).

Thanks for your attention.

---

> ### Comment · Action_Editor_5zXx · 2025-04-22
> **Final Camera Ready**
>
> Hello,
>
> Our apologies in the many delays for this review process. I noticed that in Sec. 4.5 (loss), you have an equation that is outside the sentence (period is beforehand).
>
> I believe this comment was not addressed as well:
>
> Sec 3 (Deterministic vs Stochastic Setting) - a deterministic and stochastic setting could be both multi-modal but this section seems to imply that only the deterministic setting is multi-modal. The same assumption is made in the introduction of Section 4.1.

---

### Decision · Action_Editor_5zXx · 2025-03-28

**Recommendation:** Accept with minor revision

**Comment:**

Although overall the reviewers see value in the contribution to the research community, some details and polish are still missing from the paper, hindering its reproducibility and impact. The following are comments from the reviewers that should be addressed in the final paper version.

- See Reviewer ytj4 comments on combination of different dimensions.
- Sec 3 (Deterministic vs Stochastic Setting) - a deterministic and stochastic setting could be both multi-modal but this section seems to imply that only the deterministic setting is multi-modal. The same assumption is made in the introduction of Section 4.1.
- Sec 4.4 (Decoder) - "Training" is capitalized in the middle of a sentence. "fed" --> "feed".
- Sec 4.5 (Loss) - The final sentence in this section needs to be rewritten. For example, the paper is referring to KL divergence as "the stochastic".

**Audience:**

Given that the paper focuses on getting an efficient model that performs on par with SOTA trajectory prediction architectures and supports these claims with substantial empirical results, this paper contribution seems valuable to the ML/autonomous driving community.

**Claims And Evidence:**

The paper introduces a lightweight trajectory prediction method, titled ASTRA, which has seven times fewer parameters that SOTA trajectory prediction models. The reviewers note the good overall performance and efficiency of the proposed approach. The majority of the reviewer comments were about clarity and ablations, and most of these have been resolved during the response period. As such, the empirical evidence supports the claims made in the paper.